# Tunable Octdong and Spindle-Torus Fermi Surfaces in Kramers Nodal Line Metals

Gabriele Domaine[1,2], Moritz M. Hirschmann [2,3], Kirill Parshukov [2], Mihir Date[1,4], Holger L. Meyerheim[1], Matthew D. Watson [4], Katayoon Mohseni[1], Sydney K. Y. Dufresne[1], Shigemi Terakawa[1,5,6], Marcin Rosmus [7,8], Natalia Olszowska [7], Stuart S. P. Parkin [1], Andreas P. Schnyder [2] & Niels B. M. Schröter [1,9,10] ✉

Kramers nodal lines are doubly degenerate band crossings in achiral non-centrosymmetric crystals, arising from spin-orbit coupling and connecting time-reversal invariant momenta. When intersecting the Fermi level, they generate exotic three-dimensional Fermi surfaces, in some cases described by two-dimensional massless Dirac fermions, enabling enhanced graphene-like physics such as quantized optical conductivity and large anomalous Hall effects. However, no experimental realization of such materials has been reported. Here, we identify Kramers nodal line metals beyond the case of Fermi surfaces enclosing a single time-reversal invariant momentum. Using angle-resolved photoemission spectroscopy and first-principles calculations, we show that 3R-TaS$_2$ and 3R-NbS$_2$ host open Octdong and Spindle-torus Fermi surfaces, respectively. We observe a filling-controlled transition between these configurations and evidence of size quantization in 3R-TaS$_2$ inclusions within 2H-TaS$_2$. We further predict a strain- or pressure-driven transition to a conventional metal. Our results establish 3R transition-metal dichalcogenides as a tunable platform for Kramers nodal line physics.

The discovery of graphene revolutionized condensed matter physics by unveiling exotic phenomena such as quantized optical conductivity[1] and light-induced anomalous Hall effects[2], which arise from the unique two-dimensional Dirac fermions on its Fermi surface. A key limitation of graphene, however, is the presence of only two Dirac fermions at the same energy, which significantly restricts the versatility and magnitude of the aforementioned phenomena. A major challenge in this field has thus been to extend these unique properties in materials with multiple Dirac fermions. Although the discovery of three-dimensional topological semimetals exhibiting multiple Weyl, Dirac or multi-fold fermions represented a significant milestone in that direction[3], these materials

typically do not exhibit graphene-like physics, such as quantized optical conductivity[4] due to the three-dimensional nature of their relativistic fermions. Additionally, they pose additional challenges for practical applications since the Dirac and Weyl cones can be located far from the Fermi level.

Recently, it has been predicted that all non-centrosymmetric achiral materials host so-called Kramers nodal lines (KNLs), which are doubly degenerate band crossings connecting time-reversal invariant momenta (TRIMs) in the Brillouin zone and arise due to the combination of time-reversal symmetry and achiral little group symmetries[5,6]. When Kramers nodal lines are crossing the Fermi-level

[1]Max Planck Institut für Mikrostrukturphysik, Halle, Germany. [2]Max-Planck-Institut für Festkörperforschung, Stuttgart, Germany. [3]RIKEN Center for Emergent Matter Science, Wako, Saitama, Japan. [4]Diamond Light Source Ltd, Harwell Science and Innovation Campus, Didcot, UK. [5]Department of Applied Physics, Graduate School of Engineering, The University of Osaka, Osaka, Japan. [6]Center for Future Innovation, Graduate School of Engineering, The University of Osaka, Osaka, Japan. [7]SOLARIS National Synchrotron Radiation Centre, Jagiellonian University, Krakow, Poland. [8]Université Paris-Saclay, CNRS, Institut des Sciences Moléculaires d'Orsay, Orsay, France. [9]Martin-Luther-Universität Halle-Wittenberg, Halle (Saale), Germany. [10]Halle-Berlin-Regensburg Cluster of Excellence CCE, Halle (Saale), Germany. ✉e-mail: niels.schroeter@mpi-halle.mpg.de

they form a Kramers nodal line metal (KNLM), characterized by exotic Octdong (figure-eight) and Spindle-torus Fermi-surfaces, where all electrons are described by a family of two-dimensional Dirac or Rashba Hamiltonians, respectively. Crucially, KNLM arise due to the presence of spin-orbit coupling (SOC) and have been predicted to lead to fascinating new phenomena[6] (Fig. 1a): When the materials are confined, size quantization can lead to a quantized optical conductivity with multiple quantization steps, arising from the presence of multiple Dirac cones at different energies, in contrast to the single quantized value in graphene. Furthermore, one of the Dirac cones is pinned at the Fermi level, meaning that the onset frequency for the quantization is guaranteed to be zero. This is unlike the case of generic Dirac and Weyl points in other materials, and hence lead to a finite onset frequency. Moreover, under light illumination, giant anomalous Hall effects are expected to occur due to the large number of two-dimensional massless Dirac fermions forming the Fermi-surface. Finally, since KNLs are transformed into Kramers-Weyl fermions upon breaking of the mirror or roto-inversion symmetries, KNLM are the parent state for structurally chiral Kramers-Weyl semimetals, which have so far remained elusive in experiments. As a result, KNLM could realize strain induced spin-hedgehogs[7], orbital angular momentum monopoles[8], unconventional spin-orbit torques that could be exploited for novel spintronic devices[9], and a quantized circular photogalvanic effect without multiband corrections[10]. However, due to the lack of material candidates, these theoretical predictions have so far remained untested experimentally.

It has previously been shown that if the Fermi surface encloses only one of the TRIMs connected by the KNL, the material is guaranteed to form a KNLM and the touching Fermi surfaces can be ascribed to either an octdong or spindle-torus[6]. In the former (latter), two Fermi surface pockets that enclose different TRIMs (the same TRIMs) are touching and all electrons on the Fermi surface can be described by a family of Dirac (Rashba) Hamiltonians parametrized by the momentum along the nodal line.

Although several materials have been suggested to host KNLs away from the Fermi level, such as SmAlSi[11] or transition metal ruthenium silicides[12], to the best of our knowledge, a KNLM where the KNLs are crossing the Fermi-level has not been conclusively identified in experiments yet. Whilst there have been investigations of KNLMs in YAuGe[13] and the charge density wave state of RTe₃[14,15], the crossing and splitting of the nodal line at the Fermi level is not resolved in the experiments. Moreover, although there have been predictions for the

realization of a spindle-torus Fermi surface is real materials, a candidate material hosting an exotic octdong Fermi surface that could give rise to quantized optical conductivity has not yet been identified theoretically. In this work, by combining density functional theory (DFT) and angle-resolved photoelectron spectroscopy (ARPES), we find the non-centrosymmetric rhombohedral phases (3R) of metallic $MS_2$ ($M$ = Nb, Ta) as candidates for highly tunable KNLMs, realizing both octdong and spindle-torus Fermi surfaces.

For the non-centrosymmetric space group $R3m$ (no. 160) of the 3R transition metal dichalcogenides (TMDCs), each mirror plane supports two KNLs, protected by both time-reversal symmetry and the mirror symmetry, connecting $\Gamma$ to $T$ as well as $L$ to $F$[6] (see also Section A of Supplementary Material). While the former is pinned along the high symmetry path connecting the two TRIMs, the latter is only pinned at $L$ and $F$ but is otherwise free to disperse on the mirror plane[5], hence the name almost movable KNL (AMKNL). The typical Fermi surface of a metallic 3R-TMDC, exemplified by 3R-TaS₂, together with an illustration of the nodal lines are shown in Fig. 1b, c, while the representation of the mirror plane torus, obtained by identifying the points $\mathbf{k} - \mathbf{k} + \mathbf{G}$, is shown in Fig. 1d. Although in TMDCs the Fermi surfaces typically enclose more than one single TRIM, depending on the dispersion of the AMKNL, the realization of a KNLM is still possible. The touching Fermi surfaces are then realizing an open version of the octdong (Fig. 1e) or open spindle-torous (Fig. 1f) that are described by the same type of Hamiltonians as their closed counterparts, and therefore lead to equivalent physical responses.

Interestingly, as we predict theoretically and demonstrate experimentally in this work, the Fermi surfaces of TMDCs can undergo a Lifshitz transition from octdong and spindle-torus induced by a change of the band filling, which can be also be achieved by doping or gating. In particular, our DFT calculations predict that due to the lower band filling, 3R-TaS₂ is hosting open octdong Fermi-surfaces. On the other hand, due to a higher band filling, NbS₂ realizes open spindle torus Fermi surfaces where the two touching Fermi-surfaces enclose the same TRIMs (Fig. 1c), $\Gamma$ and $T$. We confirm these predictions experimentally by directly visualizing these Fermi surfaces by ARPES.

While the 2H-phase of $MS_2$ can be synthesized in stoichiometric single crystals, the 3R-phase has been found to require doping and intercalation to be stable at ambient conditions[16,17]. This presents a formidable challenge for experimental confirmation of KNLs, because doping and intercalation can lead to broadening of experimental

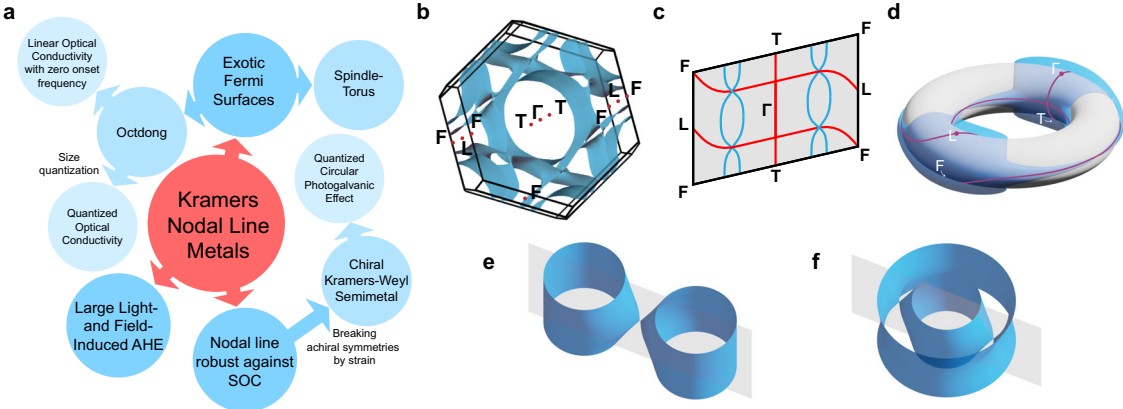

**Fig. 1 | Introducing KNLMs. a** New phenomena arising from KNLMs. **b** Example of a typical Fermi-surface for a 3R phase of TMDCs with TRIMs indicated by red dots. **c** One of the three mirror planes delimited by the high-symmetry point $F$, where an example of the KNL connectivity is shown in red, together with the points of the Fermi surface intersecting the mirror plane of 3R-TaS₂ shown in light blue. Notice that one KNL is pinned to the high symmetry direction $\Gamma - T$, while the almost-movable KNL is connecting $L$ to $F$ along an arbitrary path. **d** The quotient space of the mirror plane,

where the opposite sides of the Brillouin zone are identified, is a torus (shown in gray), while the individual Fermi surface pockets that intersect the mirror plane (shown in blue) are 2-Tori $T^2$. If the Fermi surfaces are pierced by a KNL, a touching point is enforced between the tori. Notice that, for simplicity, only the pockets intersecting the mirror plane are shown. **e** Open octdong Fermi surface, formed by an enforced touching between two pockets enclosing different TRIMs, and **f** open spindle-torus Fermi surface, with an enforced touching of two pockets enclosing the same TRIMs.

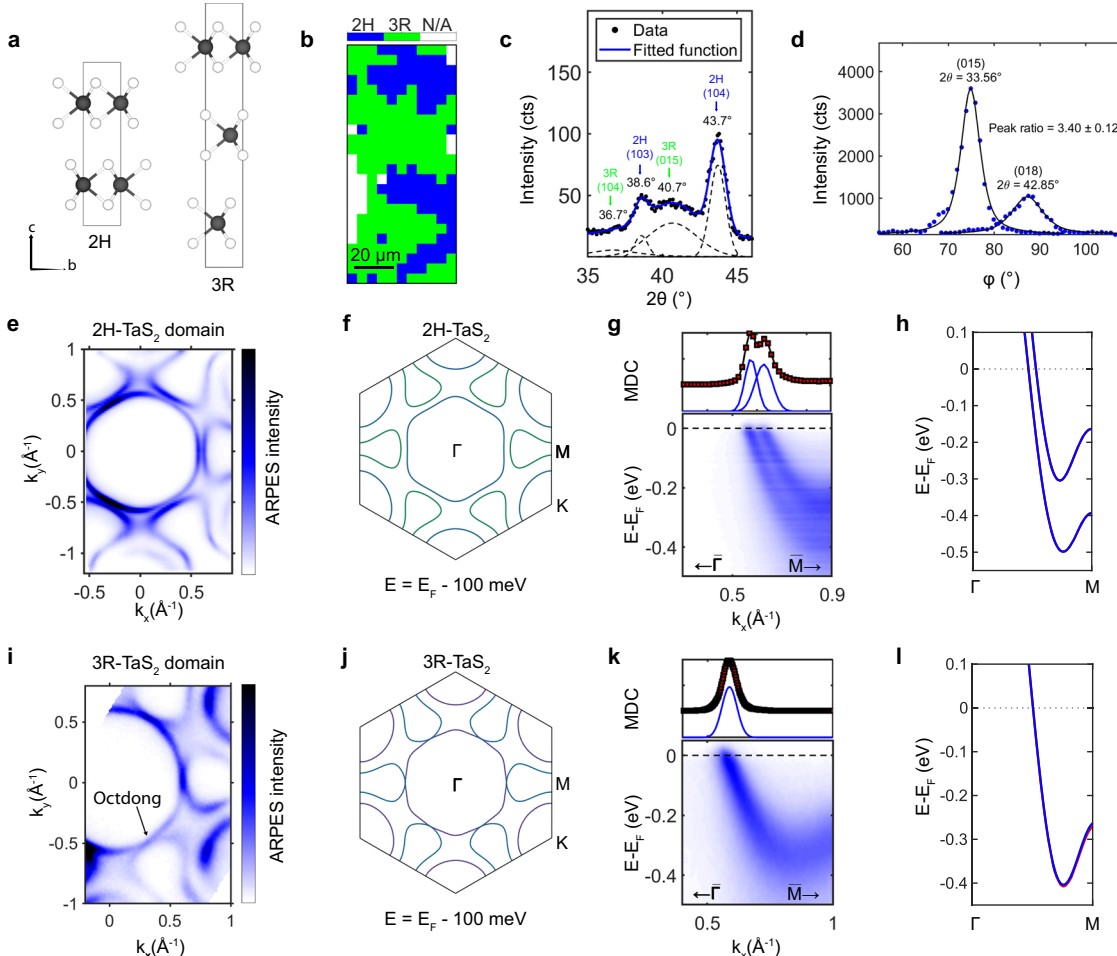

**Fig. 2 | The 2H and 3R polytypes of TaS₂.** **a** Crystal structures of the 2H and 3R polytypes of $TaS_2$ with the corresponding conventional unit cells. **b** Spatial map of a cleaved sample showing the presence of different domains identified by the number of bands crossing the Fermi level along the mirror plane ($\bar{\Gamma} - \bar{M}$ direction). **c** $2\theta - \theta$ X-ray diffraction pattern in the $2\theta = 35° - 45°$ regime showing reflections related to the $2H$-$TaS_2$ (blue) and 3R (green) polytypes. Corresponding reflection indices are given next to the reflections. Data were collected using Cu K$\alpha$ radiation ($\lambda = 1.5406$ Å). **d** Integrated transverse scans ($\varphi$) over the 3R (015) and (018) reflections, exhibiting an intensity ratio of approximately 3.4:1.0. Measurements were performed using Ga K$\alpha$ radiation ($\lambda = 1.3414$ Å). **e** Fermi surface of $2H$-$TaS_2$

measured with $hv = 50$ eV and LH (p) polarized light and **f** the corresponding Fermi surface at the $\Gamma$ plane computed by DFT. **g** ARPES data of the 2H phase along the mirror plane and **h** corresponding DFT calculation for $k_z = 0$ in blue. **i** Fermi surface of $3R$-$TaS_2$ measured with $hv = 55$ eV and LH (p) polarized light and **j** the corresponding Fermi surface at the $\Gamma$ plane computed by DFT. Here, $M$ and $K$ represent points on the rhombohedral Fermi surface at $k_z = 0$, in analogy to the hexagonal unit cell. **k** ARPES band structure of the 3R phase along the mirror plane and **l** corresponding DFT calculations for $k_z = 0$, showing the split bands in blue and red. The upper panels of **g** and **k** show the corresponding momentum distribution curves taken at the Fermi level.

ARPES spectra. However, it is well known that *as grown* single crystals of hexagonal transition metal dichalcogenides, in particular transition metal disulfides, are susceptible to stacking faults[18], which can lead to locally distinct stacking configurations and the realization of a 3R-stacking configuration without the need for additional doping. Some of the present authors have recently used microfocused ARPES to obtain relatively sharp spectra of few layer 3R-stacking configurations that were embedded within a single crystal of the 2H-polytype[17]. Here, we make use of these new capabilities to access small domains of 3R-TaS₂ with very sharp bands suggesting a low intercalation level for ARPES measurements, which enables us to confirm the presence of octdong Fermi-surfaces in this compound. We furthermore present ARPES measurements of Nb self-intercalated 3R-NbS₂, which shows that the measured Fermi-surface is consistent with a spindle-torus Fermi surface. Our DFT calculations show that one can tune between those two Fermi-surface types by changing the band filling, which can in principle be achieved by chemical doping or gating. Based on a tight-binding model, we finally demonstrate that due to the fact that the Fermi surfaces enclose multiple TRIMs, applying pressure or strain can transform these materials from Kramers-nodal line metals into

ordinary metals, where the KNL does not pierce the Fermi surface, by changing the topology of the AMKNL.

## Results

### KNLM with octdong Fermi surface in 3R-TaS₂

Because both SOC and non-centrosymmetric space groups are required for the realization of a KNLM, TMDCs involving transition metals, like Nb or Ta, are ideal candidates. During our micro-ARPES measurements of commercially available cleaved TaS₂ single crystals nominally in the 2H phase, besides the typical Fermi surface of the 2H phase, we often observed variably sized domains of another phase exhibiting one single band crossing the Fermi level along the $\Gamma - M$ direction, which we identify as the 3R phase. The crystal structures of the two phases, as well as the characteristic sizes of their respective domains, are shown in Fig. 2a, b. Figure 2c, d show the corresponding x-ray diffraction (XRD) patterns recorded in the $2\theta - \theta$ geometry. In the range $2\theta = 35° - 45°$, several reflections are observed that are uniquely attributable to either the 2H (blue) or 3R (green) TaS₂ polytypes. In particular, two characteristic 3R reflections-(015) and (018)-were recorded via transverse ($\varphi$) scans, revealing an intensity ratio of

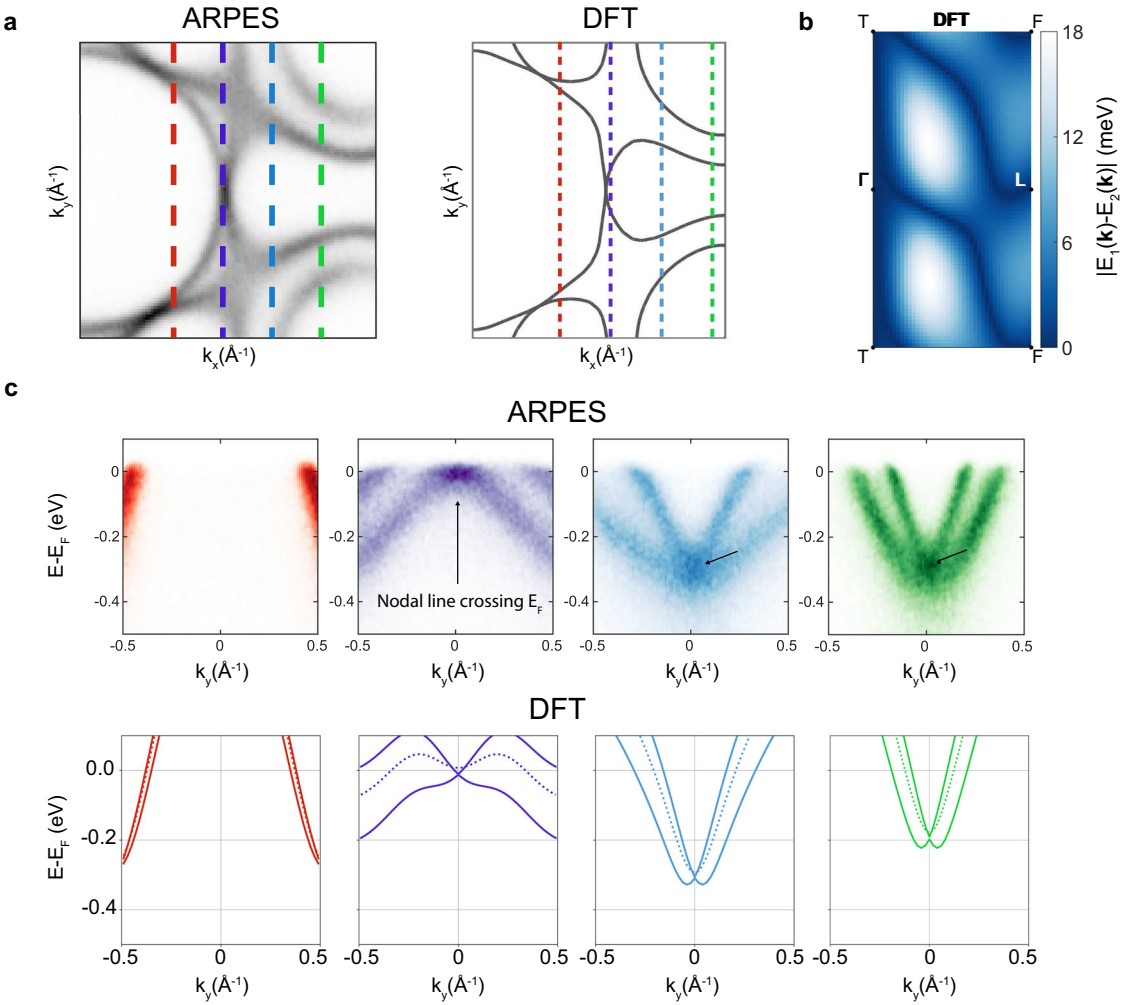

**Fig. 3 | Octdong Fermi surface formed by the nodal line piercing $E_F$ in 3R-TaS$_2$.** **a** ARPES and DFT ($k_z = 0$) Fermi surfaces showing the octdong formed by the hole and electron pockets. **b** Calculated splitting between the two bands crossing the Fermi level on the mirror plane, showing the presence of a nodal line winding twice around the mirror plane. Panel **c** show the dispersion along four different directions perpendicular to the mirror plane on both sides of the octdong touching point along the cuts in **a**, both for the ARPES data and the DFT calculations ($k_z = 0$). While going from the central hole pocket to the external electron pocket the bands on the

mirror plane move from positive to negative binding energies, meaning that the crossing point must also transition from above (red) to below (blue and green) the Fermi level, as predicted for the octdong Fermi surface in ref. 6, thus crossing the Fermi level (purple). The dotted lines represent the degenerate bands in the absence of SOC. Notice that in the DFT calculations for the green cut a small splitting of the bands on the mirror plane is visible, but this is simply due to the fact that the nodal line is not exactly at $k_z = 0$.

approximately 3.4:1.0. The positions and relative intensities of these peaks are consistent with reported values for the 3R polytype in Pearson's Crystal Data[19], which lists a ratio of approximately 3.75:1.00. Notice that the distinction between 3R, 2H, and any other possible stacking can also be made based on fundamental band theory arguments as presented in the Section B of the Supplementary Material.

The measured Fermi surface of a 2H-TaS$_2$ domain is shown in Fig. 2e, together with the corresponding density functional theory (DFT) calculation in Fig. 2f. The experimental Fermi surface shows a central hole pocket surrounded by three 'dog-bone' electron pockets along the $\bar{\Gamma} - \bar{M}$ direction (along the mirror plane). Importantly, the two pockets are not connected, as can also be seen from the experimental band dispersion along $\bar{\Gamma} - \bar{M}$ displayed in Fig. 2g, which is clearly showing two bands crossing the Fermi level. These two bands are spin-degenerate because of the presence of time-reversal- and inversion-symmetry and are mostly split by interlayer coupling, which is also confirmed by our DFT calculations shown in Fig. 2h.

In contrast, the experimental Fermi-surface of the 3R phase (Fig. 2i) shows that the two pockets are connected into a degenerate point, forming a Fermi surface shaped like an hourglass. Only a single

band is resolvable in the band dispersion along $\bar{\Gamma} - \bar{M}$ in Fig. 2k, while two bands are still visible from the experimental Fermi-surface away from the mirror plane. Our DFT calculations of 3R-TaS$_2$, shown in Fig. 2j, l, reproduce the same features observed by ARPES, strongly supporting the identification of the 2H and 3R polytypes.

This implies that the hourglass touching point along the mirror plane, if pierced by the AMKNL, can be ascribed to the open octdong formed by joining together the 'dog-bone' electron pocket to the central hole pocket. By comparing four band dispersions along a direction perpendicular to the mirror plane (Fig. 2), the experimental data clearly reveals the Kramers nodal line crossing the Fermi level in agreement with the DFT calculations. In particular, our DFT calculations also confirm that the observed splitting when moving away from the mirror plane is due to spin-orbit coupling (solid and dashed lines in the bottom panels of Fig. 3c) and also predict a nodal line that is fully winding twice around the mirror plane torus (Fig. 3b), implying that it must pierce the Fermi surface. The Dirac-like dispersion that is shifting through the Fermi level when going from touching hole to electron pockets (Fig. 3a, c) is the predicted hallmark of the octdong Fermi surface of KNLMs and has never been observed in experiments

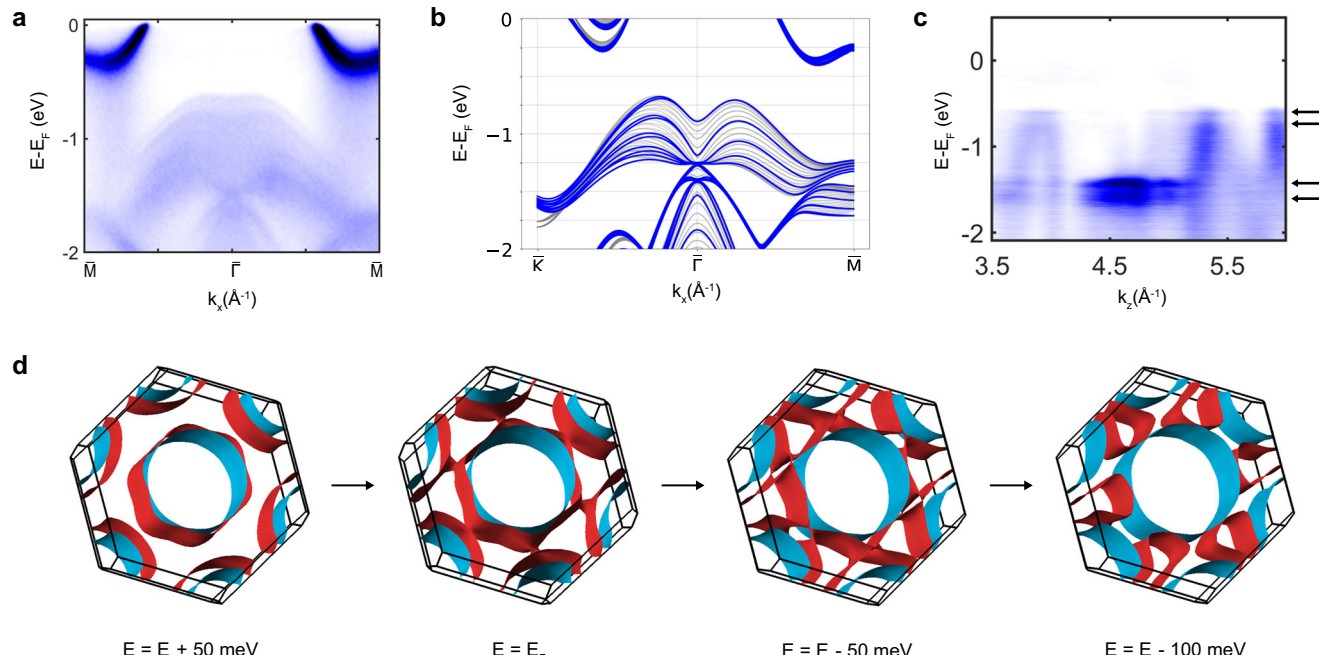

**Fig. 4 | Quantum-well states observed in 3R-TaS₂ and Fermi-surface transition with band filling. a** Band structure of 3R-TaS$_2$ at 55 eV on the mirror plane, showing a quantization of the valence band and **b** the corresponding 5-layers slab (blue) and bulk (gray) calculation obtained from DFT. **c** Quasi-two-dimensional character of some of the bands at $k_x = 0$ as shown by their flat dispersion as a function of the out-of-plane momentum $k_z$. **d** Change in the topology of isoenergy surfaces of 3R TaS$_2$ upon change of the binding energy, with the two bands represented in blue and red. As the binding energy is reduced, the system transitions from concentric pockets to pockets surrounding different TRIMs.

(c.f. Fig. 3e in ref. 6). Note that the gapless nodal line only occurs at specific out-of-plane momenta $k_z$ within the mirror plane, which would technically correspond to a specific photon energy in an ARPES experiment. However, due to the limited probing depth of our experiment that results in out-of-plane momentum broadening[20], we observe a gapless nodal line spectrum for all photon energies within our experimental resolution. Furthermore, here we only show the first Brillouin zone, but as shown in the Section C of the Supplementary Material, the touching point between the two pockets is reproduced also in neighboring Brillouin zones.

**Naturally occurring quantum confinement in 3R-TaS₂ samples**

Identifying a naturally occuring quantum confinement of the 3R-TaS$_2$ within the 2H-TaS$_2$ sample would be particularly appealing since the octdong Fermi surface can be described by a two-dimensional Dirac Hamiltonian, which leads to a quantization of the optical conductivity when the momentum along the nodal line is quantized. This was originally proposed by considering size quantization in the thin film limit[6]. Intriguingly, our ARPES measurements of the 3R-TaS$_2$ domains show signatures of quantum well states in the valence band (Fig. 4a–c), similar to the 3R-NbS$_2$ and 3R-MoS$_2$ thin film inclusions that were recently reported in commercial single crystals nominally of the 2H-polytype[21]. This observation suggests that the 3R phase on the surface of the commercially available 2H-TaS$_2$ could consist of only a few layers, which would also lead to a quantization of the out-of-plane momentum, implying that it could directly exhibit a quantization of the optical conductivity. Based on the number of quantized bands, we estimate that the 3R phase should be approximately 5-monolayers thick. This value is also supported by the analysis of the width of the fitted XRD peaks (see the Section D of the Supplementary Material). To get a reliable estimate of the bulk 2H- vs. 3R-phase volume ratio in the crystal, we performed additional quantitative single crystal X ray diffraction experiments which allowed us to estimate a bulk volume ratio of approximately 8:1 in favor of the 2H phase (see the Section D of the Supplementary Material). This partially deviates from the ARPES

data, which show nearly equal surface contributions of 2H and 3R. We attribute the enhanced presence of the 3R phase at the surface either to preferential cleaving at 2H and 3R stacking faults or to cleaving induced stacking faults[22].

An alternative explanation of our data could be surface band bending, which has also been invoked to explain quantum well states in the valence band of TaSe$_2$[23], due to a confining surface potential. Such band-bending induced surface confinement would also lead to a quantization of the conduction band and thus quantized optical conductivity similar to the thin film case, which could be investigated in future optical experiments. A third explanation could be that the observed valence band features are actually genuine surface states or surface resonances rather than quantum well states, in which case the conduction band would not have to be quantized. Finally, beyond the well-known quantum well confinement at III-V semiconductor phase boundaries—exploited in optoelectronic devices[24]—similar confinement potentials are also known to arise in metallic quantum wells because of relative and hybridization gaps[25].

**Tuning the octdong into a spindle-torus by increased band filling in 3R-TMDCs**

Another interesting feature of the 3R-TMDCs is that, because of their characteristic band structure along the $\bar{\Gamma} - \bar{K}$ path, increasing the filling factor can lead to a Lifshitz transition from the open octdong to the open spindle-torus configuration, as shown by the DFT calculation comparing isoenergy surfaces at different binding energy of 3R-TaS$_2$ in Fig. 4d. Since only the octdong Fermi surface is expected to show quantized optical conductivity, doping or gating these compounds could be used to significantly tune the optoelectronic properties of these materials. To demonstrate this tunability with band filling, we also investigated the commercially available 3R-NbS$_2$ polytype, which is known to have an increased band-filling due to self-doping by Nb interstitials during growth[17]. This material is predicted by DFT to exhibit two touching concentric Fermi surfaces centered at $\Gamma$ (Fig. 5a), surrounded by six disconnected Fermi surface pockets that are split

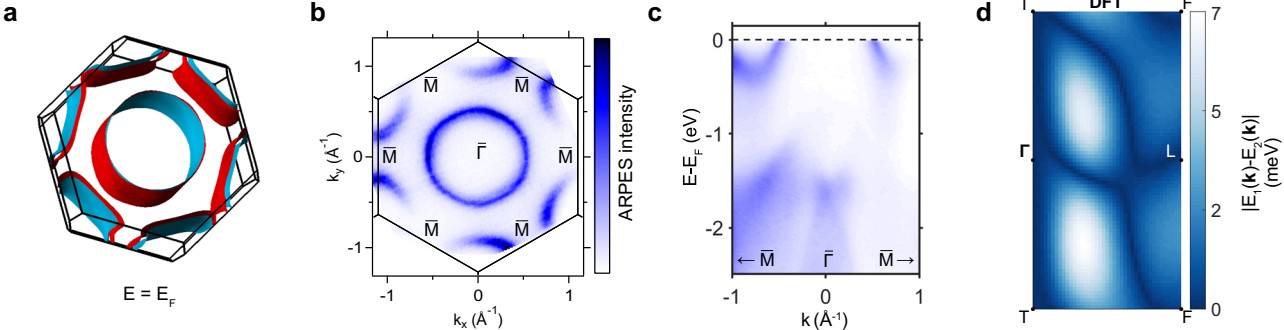

**Fig. 5 | Spindle-torus Fermi surface in 3R-NbS₂. a** Three-dimensional Fermi surface of 3R-NbS₂ computed by DFT. **b** 70-eV ARPES Fermi surface of 3R-NbS₂ and **c** band dispersion along the high-symmetry path $\bar{\Gamma} - \bar{M}$. **d** Magnitude of the splitting between the two bands crossing the Fermi level on the mirror plane, showing the AMKNL connecting $L$ to $F$ by winding around the mirror plane torus.

due to spin-orbit coupling. The predicted Fermi surface is also confirmed by ARPES, showing a central hole pocket surrounded by six disconnected pockets (Fig. 5b). Since the spin-orbit coupling in this compound is relatively small and the spectral features are broad, likely due to the presence of interstitial Nb atoms[17], we were not able to resolve the predicted spin-orbit splitting of the bands in our ARPES measurement. However, our DFT calculations predict an almost movable nodal line that fully winds twice around the mirror plane torus, so that it is guaranteed to cross the Fermi surface at four points. This would imply that the two hole pockets enclosing $\Gamma$ and $T$ realize an open spindle-torus Fermi surface, which is consistent with our ARPES measurements.

### Nodal line topology and its tunability by strain

As already mentioned in the previous sections, while the existence of the nodal lines is symmetry-enforced, the dispersion of the AMKNL is material-dependent so that one can make a distinction between those that are guaranteed to cross the Fermi surface and those that are not. To do so, one can start by realizing that the quotient space of the mirror plane with the prescription $\mathbf{k} \sim \mathbf{k} + \mathbf{G}$, where $\mathbf{G}$ is a reciprocal lattice vector, is a torus $T^2$. On the other hand, both the mirror-invariant points of the Fermi surface and the AMKNL connecting $L$ and $F$ can be ascribed to one-dimensional loops $S^1 \subset T^2$. The different mappings $f$ of loops onto $T^2$ are elements of the fundamental homotopy group $\pi_1(T^2) = \mathbb{Z} \times \mathbb{Z}$ where each class is identified by a set of two winding numbers, respectively along the meridian and longitude, $f \equiv [w_m, w_l] \in \pi_1(T^2)$. If we take the high-symmetry path $\Gamma - T$ to lie along the torus' meridian, then we have that the Fermi surface loops and the pinned KNL connecting $\Gamma$ and $T$ both belong to the classes $[\pm 1, 0]$. Because for symmorphic mirror symmetries there can only be an odd number of lines originating from each TRIM due to TRS exchanging the mirror symmetry eigenvalues, the AMKNL is restrained to the classes $[2\mathbb{Z}+1, 2\mathbb{Z}]$. Then, if $f_{AMKNL} \in [1, w_l \neq 0]$, each Fermi surface loop must be pierced by the nodal line at least $w_l$ times. On the other hand, if $f_{AMKNL} \in [w_m, 0]$ the crossing is not guaranteed and will depend on the relative position of the nodal line and the Fermi surface. This means that there can be two types of KNLM, those with $w_l \neq 0$ where the Fermi surface crossing is guaranteed, and those with $w_l = 0$, which are accidental (Fig. 6). Because each $w_l$ identifies topologically distinct nodal lines, a transition from one class to the other must happen through a Lifshitz transition of the AMKNL. For both NbS₂ and TaS₂ the DFT calculations predict $w_l = 2$, however we expect it to be possible to enforce a transition to $w_l = 0$ by applying strain to the system to affect the out-of-plane dispersion of the nodal line.

To provide an intuitive understanding of this, we construct a tight-binding model with a single orbital and a spin-1/2 degree of freedom. The model accounts for in-plane terms up to fifth nearest-neighbor and out-of-plane terms up to third nearest-neighbor resulting in a Hamiltonian consisting of a sum of 17 matrix components and corresponding hopping terms. In order to model the effects of uniaxial strain, all hopping parameters involving any out-of-plane displacement are scaled, thereby emulating the modification of orbital overlaps. Importantly, the spin-orbit splitting within the mirror planes, and thus the shape of the nodal lines, is fully determined by the relative sizes of out-of-plane and in-plane hopping terms. A dominant in-plane hopping results in nodal lines that close the spin-orbit gap when traversing the Brillouin zone in-plane, as seen in Fig. 6a. Conversely, when out-of-plane spin-orbit terms dominate, the spin-orbit gap closes along the vertical direction, and the nodal line crosses the $\Gamma - T$ path (Fig. 6b). Strain-induced tuning of these terms can thus drive a transition in the nodal line connectivity. Details of the full tight-binding Hamiltonian are provided in Methods, and comparison with the DFT band structure is shown in the Section E of the Supplementary Material.

## Discussion

We have demonstrated using ARPES and theoretical calculations that the 3R polytypes of TaS₂ and NbS₂ are KNLMs, exhibiting, respectively, octdong and spindle torus Fermi surfaces enforced by a KNL crossing the Fermi level. We have furthermore demonstrated that Fermi surfaces that enclose multiple TRIMs allow for high tunability between octdong and spindle-torus by band filling, as well as topological KNLM to trivial metal transition by strain. Future experiments could test the quantized optical conductivity of confined samples with octdong Fermi surface of 3R-TaS₂, which we have demonstrated to naturally occur on the surface of as grown 2H-TaS₂ due to stacking faults or surface band bending. Moreover, the gate-tunability of these systems could even allow for junctions between octdong and spindle-torus Fermi surfaces that could be used to investigate the scattering of Dirac- into Rashba-electrons, and vice-versa. These findings establish the 3R polytypes of metallic TMDCs as an ideal tunable platform for studying new phenomena predicted for KNLMs. Our detailed single-crystal X-ray diffraction experiments revealed a small amount of disorder in our samples where about 10(5)% Ta atoms from the main octahedral site are redistributed into interstitial positions. Such self-intercalation effects have been reported to favor the 3R-polytype in other TMDCs such as NbS₂[26], and samples with excess Ta have been shown to form 3R-Ta₁.₀₈S₂ with Ta occupying the same interstitial positions as found in our study[16]. Possible replication strategies may therefore have to source samples from the same supplier or to develop alternative growth recipes that favors the occupation of interstitial sites while maintaining stoichiometry, or alternatively study TaS₂ samples that were grown with an excess of Ta to form the 3R-polytype.

## Methods
### ARPES
Single crystals of 3R-NbS₂ and 2H-TaS₂ were obtained commercially from HQ Graphene, Groningen. All crystals were used as received from

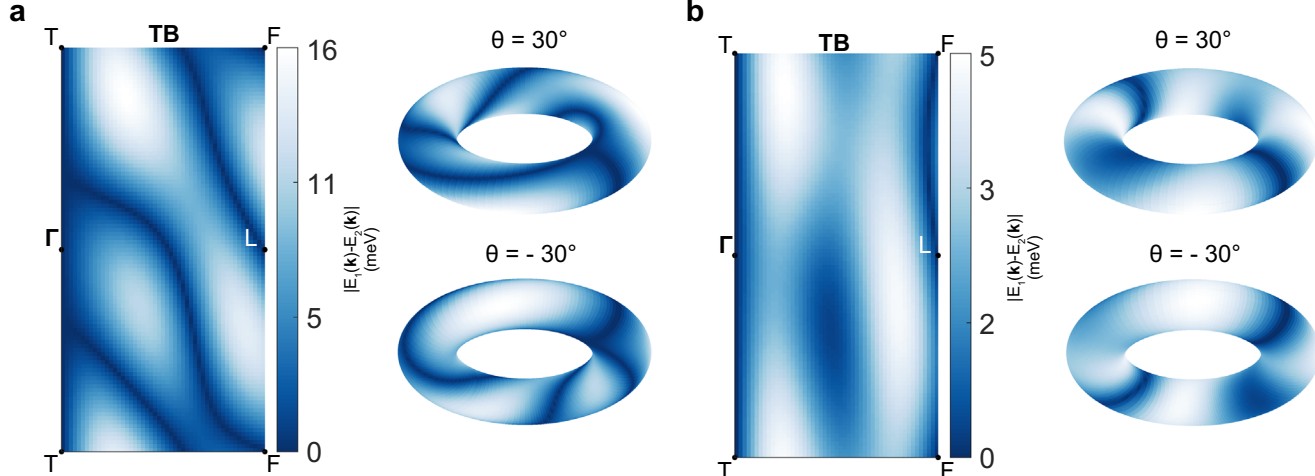

**Fig. 6 | Tunability of nodal line connectivity with strain. a, b** Splitting between the two bands crossing the Fermi level showing two different types of connectivity. In **a** the parameters of the tight-binding model are the ones obtained from fitting the model to the bands computed by DFT. On the other hand in **b**, to capture the effect of a lattice deformation, the out-of-plane hopping amplitudes are multiplied by a factor of 1/10, see Methods Sec. "Tight binding model" for details. The figures also show the mapping of the nodal lines onto the mirror plane torus (respectively viewed from an elevation of 30° and −30°), where it is apparent that the config-uration with the reduced out-of-plane hopping does not wind around the torus, and as such is in general not guaranteed to cross the Fermi level. On the other hand, the configuration with a stronger out-of-plane hopping fully winds twice around the torus, and is thus guaranteed to pierce the Fermi surface. Notice also that the nodal line winding along the meridian on the right-hand side of the torus corresponds to the KNL pinned along the high-symmetry path $\Gamma - T$.

HQ Graphene and were not subjected to any chemical, thermal, or mechanical treatment beyond standard handling for XRD and ARPES measurements. The presence of Ta interstitials revealed by our XRD refinements (see Section D of the Supplementary Material) provides a plausible growth-related mechanism for stabilizing the 3R-TaS$_2$ domains, consistent with previous literature[16]. ARPES measurements of TaS$_2$ were performed at the I05 beamline of Diamond Light Source[27], with a Scienta DA-30 analyser measaured at a pressure of approx. 1-2 × 10$^{-10}$ mbar, and using photon energies in the range 30-150 eV, at sample temperatures between 80 and 100 K. The combined energy resolution was approximately 30-40 meV with an approximate beam spot diameter (FWHM) of 4 $\mu$m. On the other hand, the measurements of NbS$_2$ were performed at the URANOS beamline at the National Synchrotron Radiation Centre SOLARIS in Krakow (Poland) using a SCIENTA OMICRON DA30L photoelectron spectrometer, at the sample temperature of 15 K. The photon energy used ranged from 20-100 eV.

## DFT

Band structures and Fermi surfaces of (2H, 3R)-TaS$_2$ and 3R-NbS$_2$ were obtained with DFT. We used Perdew-Burke-Ernzerhof functional implemented in the Vienna Ab Initio Simulation Package (VASP)[28–31] utilizing projector-augmented wave method[32,33] with kinetic energy cutoff for the plane-wave basis 520 eV. Both 3R phases are described by symmetries of SG 160 (R3m) with atoms located at 3a Wyckoff position. Optimized positions of atoms for 3R-NbS$_2$ were taken from Materials Project[34] with lattice parameters $a = b = c = 7.087$ Å, $\alpha = \beta = \gamma = 27.403°$, the compound has the ID mp-966 (ICSD 42099[35]). For the calculation of 2H-TaS$_2$ (SG 194) electronic structure, we relaxed a hexagonal cell with lattice parameters $a = b = 3.342$ Å, $c = 13.760$ Å, $\alpha = \beta = 90°$, $\Gamma = 120°$ until the total energy change between two steps was less than $1e^{-7}$. The compound has the IDs mp-1984 and ICSD 68488. Relaxed positions of atoms for 3R-TaS$_2$ were taken for the compound with the code mp-10014 (ICSD 43410), the lattice parameters were $a = b = c = 7.089$ Å, $\alpha = \beta = \gamma = 27.281°$. To compute the Fermi surface we performed the calculations on a k-grid 41 × 41 × 41. For the slab calculations of 3R-TaS$_2$, we constructed the supercell with 5 monolayers and relaxed it until the total energy change between the two steps was less than $1e^{-4}$.

## Tight binding model

The complete Hamiltonian is expressed as

$$H(\boldsymbol{k}) = \sum_{n=1}^{17} H_n(\boldsymbol{k}), \qquad (1)$$

where, using the primitive lattice vectors $\boldsymbol{a}_1, \boldsymbol{a}_2, \boldsymbol{a}_3$ and a spin-1/2 basis, the matrices $H_n(\boldsymbol{k})$ comprise hopping terms in-plane up to 5th NN and out-of-plane up to 3rd order hopping terms.

$$H_1(\boldsymbol{k}) = 2\Re(t_1)1_{2\times2} \qquad (2)$$

$$(H_2(\boldsymbol{k}))_{12} = 2i\left(t_2 + t_2^* e^{\frac{i\pi}{3}}\right)\left(e^{\frac{i2\pi}{3}}\sin(\boldsymbol{k}\cdot\boldsymbol{a}_1) + \sin(\boldsymbol{k}\cdot\boldsymbol{a}_2) + e^{-\frac{i2\pi}{3}}\sin(\boldsymbol{k}\cdot\boldsymbol{a}_3)\right) \qquad (3)$$

$$(H_3(\boldsymbol{k}))_{12} = 2i\left(t_3 + t_3^* e^{\frac{i2\pi}{3}}\right)\left(e^{\frac{i2\pi}{3}}\sin(\boldsymbol{k}\cdot(\boldsymbol{a}_1 - \boldsymbol{a}_2)) + \sin(\boldsymbol{k}\cdot(\boldsymbol{a}_2 - \boldsymbol{a}_3))\right.$$
$$\left. + e^{-\frac{i2\pi}{3}}\sin(\boldsymbol{k}\cdot(\boldsymbol{a}_3 - \boldsymbol{a}_1))\right) \qquad (4)$$

$$(H_4(\boldsymbol{k}))_{11} = 4\Re(t_4)\left(\cos(\boldsymbol{k}\cdot\boldsymbol{a}_1) + \cos(\boldsymbol{k}\cdot\boldsymbol{a}_2) + \cos(\boldsymbol{k}\cdot\boldsymbol{a}_3)\right) \qquad (5)$$

$$(H_5(\boldsymbol{k}))_{11} = 2\Re(t_5)\left(\cos(\boldsymbol{k}\cdot(\boldsymbol{a}_1 - \boldsymbol{a}_2)) + \cos(\boldsymbol{k}\cdot(\boldsymbol{a}_2 - \boldsymbol{a}_3)) + \cos(\boldsymbol{k}\cdot(\boldsymbol{a}_3 - \boldsymbol{a}_1))\right)$$
$$+ 2\Im(t_5)\left(\sin(\boldsymbol{k}\cdot(\boldsymbol{a}_1 - \boldsymbol{a}_2)) + \sin(\boldsymbol{k}\cdot(\boldsymbol{a}_2 - \boldsymbol{a}_3)) + \sin(\boldsymbol{k}\cdot(\boldsymbol{a}_3 - \boldsymbol{a}_1))\right) \qquad (6)$$

$$(H_6(\boldsymbol{k}))_{12} = 4\Im(t_6)\left(e^{-\frac{i\pi}{3}}\sin(\boldsymbol{k}\cdot(\boldsymbol{a}_1 - \boldsymbol{a}_2 + \boldsymbol{a}_3)) - \sin(\boldsymbol{k}\cdot(\boldsymbol{a}_1 + \boldsymbol{a}_2 - \boldsymbol{a}_3))\right.$$
$$\left. + e^{\frac{i\pi}{3}}\sin(\boldsymbol{k}\cdot(-\boldsymbol{a}_1 + \boldsymbol{a}_2 + \boldsymbol{a}_3))\right) \qquad (7)$$

$$(H_7(\boldsymbol{k}))_{11} = 4\Re(t_7)\left(\cos(\boldsymbol{k}\cdot(-\boldsymbol{a}_1 + \boldsymbol{a}_2 + \boldsymbol{a}_3)) + \cos(\boldsymbol{k}\cdot(\boldsymbol{a}_1 - \boldsymbol{a}_2 + \boldsymbol{a}_3))\right.$$
$$\left. + \cos(\boldsymbol{k}\cdot(\boldsymbol{a}_1 + \boldsymbol{a}_2 - \boldsymbol{a}_3))\right) \qquad (8)$$

**Table 1 | Tight-binding parameters in Eq. (1) for 3R-TaS$_2$ in units of meV**

| | | | | | |
|---|---|---|---|---|---|
| $t_1 = 1877$ | $t_2 = 0.831$ | $t_3 = 0.662$ | $t_4 = -7.68$ | $t_5 = 75.8 + 26.6i$ | $t_6 = -0.505i$ |
| $t_7 = 2.52$ | $t_8 = -0.116$ | $t_9 = 57.7$ | $t_{10} = -0.00279i$ | $t_{11} = -1.17 - 0.0921i$ | $t_{12} = -2.69$ |
| $t_{13} = -17.3 - 0.923i$ | $t_{14} = -0.232$ | $t_{15} = 0.459$ | $t_{16} = 0.077 + 0.29i$ | $t_{17} = -13.0 + 2.83i$ | |

$$(H_8(\boldsymbol{k}))_{12} = 2i\left(t_8 + t_8^* e^{\frac{i\pi}{3}}\right)\left(e^{\frac{i2\pi}{3}}\sin(\boldsymbol{k}\cdot(2\boldsymbol{a}_1 - \boldsymbol{a}_2 - \boldsymbol{a}_3)) + \sin(\boldsymbol{k}\cdot(-\boldsymbol{a}_1 + 2\boldsymbol{a}_2 - \boldsymbol{a}_3))\right.$$
$$\left. + e^{-\frac{i2\pi}{3}}\sin(\boldsymbol{k}\cdot(-\boldsymbol{a}_1 - \boldsymbol{a}_2 + 2\boldsymbol{a}_3))\right)$$
$$(9)$$

$$(H_9(\boldsymbol{k}))_{11} = 4\Re(t_9)\left(\cos(\boldsymbol{k}\cdot(2\boldsymbol{a}_1 - \boldsymbol{a}_2 - \boldsymbol{a}_3)) + \cos(\boldsymbol{k}\cdot(-\boldsymbol{a}_1 + 2\boldsymbol{a}_2 - \boldsymbol{a}_3))\right.$$
$$\left. + \cos(\boldsymbol{k}\cdot(-\boldsymbol{a}_1 - \boldsymbol{a}_2 + 2\boldsymbol{a}_3))\right)$$
$$(10)$$

$$(H_{10}(\boldsymbol{k}))_{12} = 4\Im(t_{10})\left(-\sin(\boldsymbol{k}\cdot(\boldsymbol{a}_1 + \boldsymbol{a}_2)) + e^{\frac{i\pi}{3}}\sin(\boldsymbol{k}\cdot(\boldsymbol{a}_2 + \boldsymbol{a}_3))\right.$$
$$\left. + e^{-\frac{i\pi}{3}}\sin(\boldsymbol{k}\cdot(\boldsymbol{a}_3 + \boldsymbol{a}_1))\right)$$
$$(11)$$

$$(H_{11}(\boldsymbol{k}))_{12} = 2it_{11}\left(e^{\frac{i2\pi}{3}}\sin(\boldsymbol{k}\cdot(2\boldsymbol{a}_1 - \boldsymbol{a}_2)) + \sin(\boldsymbol{k}\cdot(2\boldsymbol{a}_2 - \boldsymbol{a}_3)) + e^{-\frac{i2\pi}{3}}\sin(\boldsymbol{k}\cdot(2\boldsymbol{a}_3 - \boldsymbol{a}_1))\right)$$
$$+ 2it_{11}^*\left(e^{\frac{i\pi}{3}}\sin(\boldsymbol{k}\cdot(2\boldsymbol{a}_2 - \boldsymbol{a}_1)) - \sin(\boldsymbol{k}\cdot(2\boldsymbol{a}_1 - \boldsymbol{a}_3)) + e^{-\frac{i\pi}{3}}\sin(\boldsymbol{k}\cdot(2\boldsymbol{a}_3 - \boldsymbol{a}_2))\right)$$
$$(12)$$

$$(H_{12}(\boldsymbol{k}))_{11} = 4\Re(t_{12})(\cos(\boldsymbol{k}\cdot(\boldsymbol{a}_1 + \boldsymbol{a}_2)) + \cos(\boldsymbol{k}\cdot(\boldsymbol{a}_2 + \boldsymbol{a}_3)) + \cos(\boldsymbol{k}\cdot(\boldsymbol{a}_1 + \boldsymbol{a}_3)))$$
$$(13)$$

$$(H_{13}(\boldsymbol{k}))_{11} = 2\Re(t_{13})\left(\cos(\boldsymbol{k}\cdot(\boldsymbol{a}_1 - 2\boldsymbol{a}_2)) + \cos(\boldsymbol{k}\cdot(\boldsymbol{a}_2 - 2\boldsymbol{a}_3)) + \cos(\boldsymbol{k}\cdot(\boldsymbol{a}_3 - 2\boldsymbol{a}_1))\right.$$
$$+ \cos(\boldsymbol{k}\cdot(2\boldsymbol{a}_3 - \boldsymbol{a}_1)) + \cos(\boldsymbol{k}\cdot(2\boldsymbol{a}_1 - \boldsymbol{a}_2)) + \cos(\boldsymbol{k}\cdot(2\boldsymbol{a}_2 - \boldsymbol{a}_3)))$$
$$+ 2\Im(t_{13})(\sin(\boldsymbol{k}\cdot(\boldsymbol{a}_1 - 2\boldsymbol{a}_2)) + \sin(\boldsymbol{k}\cdot(\boldsymbol{a}_2 - 2\boldsymbol{a}_3)) + \sin(\boldsymbol{k}\cdot(\boldsymbol{a}_3 - 2\boldsymbol{a}_1))$$
$$\left. + \sin(\boldsymbol{k}\cdot(2\boldsymbol{a}_3 - \boldsymbol{a}_1)) + \sin(\boldsymbol{k}\cdot(2\boldsymbol{a}_1 - \boldsymbol{a}_2)) + \sin(\boldsymbol{k}\cdot(2\boldsymbol{a}_2 - \boldsymbol{a}_3)))\right)$$
$$(14)$$

$$(H_{14}(\boldsymbol{k}))_{12} = 2i\left(t_{14} + t_{14}^* e^{\frac{i2\pi}{3}}\right)\left(e^{\frac{i2\pi}{3}}\sin(\boldsymbol{k}\cdot 2(\boldsymbol{a}_1 - \boldsymbol{a}_2)) + \sin(\boldsymbol{k}\cdot 2(\boldsymbol{a}_2 - \boldsymbol{a}_3))\right.$$
$$\left. + e^{-\frac{i2\pi}{3}}\sin(\boldsymbol{k}\cdot 2(\boldsymbol{a}_3 - \boldsymbol{a}_1))\right)$$
$$(15)$$

$$(H_{15}(\boldsymbol{k}))_{11} = 2\Re(t_{15})(\cos(\boldsymbol{k}\cdot 2(\boldsymbol{a}_1 - \boldsymbol{a}_2)) + \cos(\boldsymbol{k}\cdot 2(\boldsymbol{a}_2 - \boldsymbol{a}_3)) + \cos(\boldsymbol{k}\cdot 2(\boldsymbol{a}_3 - \boldsymbol{a}_1)))$$
$$+ 2\Im(t_{15})(\sin(\boldsymbol{k}\cdot 2(\boldsymbol{a}_1 - \boldsymbol{a}_2)) + \sin(\boldsymbol{k}\cdot 2(\boldsymbol{a}_2 - \boldsymbol{a}_3)) + \sin(\boldsymbol{k}\cdot 2(\boldsymbol{a}_3 - \boldsymbol{a}_1)))$$
$$(16)$$

$$(H_{16}(\boldsymbol{k}))_{12} = 2it_{16}\left(e^{\frac{i\pi}{3}}\sin(\boldsymbol{k}\cdot(2\boldsymbol{a}_1 + \boldsymbol{a}_2 - 3\boldsymbol{a}_3)) + e^{-\frac{i\pi}{3}}\sin(\boldsymbol{k}\cdot(-3\boldsymbol{a}_1 + 2\boldsymbol{a}_2 + \boldsymbol{a}_3))\right.$$
$$\left. - \sin(\boldsymbol{k}\cdot(\boldsymbol{a}_1 - 3\boldsymbol{a}_2 + 2\boldsymbol{a}_3))\right)$$
$$+ 2it_{16}^*\left(e^{\frac{i2\pi}{3}}\sin(\boldsymbol{k}\cdot(\boldsymbol{a}_1 + 2\boldsymbol{a}_2 - 3\boldsymbol{a}_3)) + \sin(\boldsymbol{k}\cdot(-3\boldsymbol{a}_1 + \boldsymbol{a}_2 + 2\boldsymbol{a}_3))\right.$$
$$\left. + e^{-\frac{i2\pi}{3}}\sin(\boldsymbol{k}\cdot(2\boldsymbol{a}_1 - 3\boldsymbol{a}_2 + \boldsymbol{a}_3))\right)$$
$$(17)$$

$$(H_{17}(\boldsymbol{k}))_{11} = 2\Re(t_{17})\left(\cos(\boldsymbol{k}\cdot(3\boldsymbol{a}_1 - 2\boldsymbol{a}_2 - \boldsymbol{a}_3)) + \cos(\boldsymbol{k}\cdot(-3\boldsymbol{a}_1 + \boldsymbol{a}_2 + 2\boldsymbol{a}_3))\right.$$
$$+ \cos(\boldsymbol{k}\cdot(-\boldsymbol{a}_1 + 3\boldsymbol{a}_2 - 2\boldsymbol{a}_3)) + \cos(\boldsymbol{k}\cdot(2\boldsymbol{a}_1 - 3\boldsymbol{a}_2 + \boldsymbol{a}_3))$$
$$+ \cos(\boldsymbol{k}\cdot(-2\boldsymbol{a}_1 - \boldsymbol{a}_2 + 3\boldsymbol{a}_3)) + \cos(\boldsymbol{k}\cdot(\boldsymbol{a}_1 + 2\boldsymbol{a}_2 - 3\boldsymbol{a}_3)))$$
$$+ 2\Im(t_{17})(\sin(\boldsymbol{k}\cdot(3\boldsymbol{a}_1 - 2\boldsymbol{a}_2 - \boldsymbol{a}_3)) + \sin(\boldsymbol{k}\cdot(-3\boldsymbol{a}_1 + \boldsymbol{a}_2 + 2\boldsymbol{a}_3))$$
$$+ \sin(\boldsymbol{k}\cdot(-\boldsymbol{a}_1 + 3\boldsymbol{a}_2 - 2\boldsymbol{a}_3)) + \sin(\boldsymbol{k}\cdot(2\boldsymbol{a}_1 - 3\boldsymbol{a}_2 + \boldsymbol{a}_3))$$
$$\left. + \sin(\boldsymbol{k}\cdot(-2\boldsymbol{a}_1 - \boldsymbol{a}_2 + 3\boldsymbol{a}_3)) + \sin(\boldsymbol{k}\cdot(\boldsymbol{a}_1 + 2\boldsymbol{a}_2 - 3\boldsymbol{a}_3)))\right)$$
$$(18)$$

The remaining entries on the diagonal and off-diagonal are given by $(H_n(\boldsymbol{k}))_{22} = (H_n(-\boldsymbol{k}))_{11}$ and $(H_n(\boldsymbol{k}))_{21} = (H_n(\boldsymbol{k}))_{12}^*$, respectively. All not specified entries in the matrices $H_n(\boldsymbol{k})$ are zero.

The parameters $t_1$ to $t_{17}$ are adjusted to match the DFT band structure of 3R-TaS$_2$ on the path S-$\Gamma$-L that are evenly spaced in $k_z$ between $\Gamma$ and $T$, resulting in Table 1. We approximate the effect of c-axis pressure by multiplying all out-of-plane hopping terms, $t_2, t_4, t_6, t_7, t_{10}, t_{11}, t_{12}, t_{13}$, by a common factor. The length of the out-of-plane component of all relevant hopping vectors are the same, hence we include only one factor for all out-of-plane terms.

## Data availability
The primary data generated in this study have been deposited in the Open Research Data Repository of the Max Planck Society under accession code https://doi.org/10.17617/3.LRVAXK, see ref. 36.

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

## Acknowledgements

G.D. and S.K.Y.D acknowledge proposal SI39232 at the i05-1 Endstation at the Diamond Light Source, UK. G.D. and S. T. acknowledge proposal 20227155 at the URANOS beamline at the National Synchrotron Radiation Centre "SOLARIS", Poland. This publication was partially developed under the provision of the Polish Ministry of Science and Higher Education project 'Support for research and development with the use of research infrastructure of the National Synchrotron Radiation Centre SOLARIS' under contract no 1/SOL/2021/2. N.B.M.S. acknowledges funding by the European Union (ERC Starting Grant ChiralTopMat, Project No. 101117424). M.M.H. is funded by the Deutsche Forschungsgemeinschaft (DFG, German Research Foundation) - project number 518238332. K.P. and A.P.S. are funded by the Deutsche Forschungsgemeinschaft (DFG, German Research Foundation) - TRR 360 - 492547816. G.D. acknowledges support by the Max Planck Graduate Center for Quantum Materials (MPGC-QM). G.D. and M.D. acknowledge Jenny Davern for her assistance in conceptualizing Fig. 1a.

## Author contributions

G.D., M.D.W., and M.D. performed the ARPES measurements with support from S.K.Y.D., S.T., M.R., and N.O. M.M.H. performed the tight-binding modeling and K.P. performed the DFT calculations. H.L.M. and K.M. performed the XRD measurements and analysis. S.S.P.P., A.P.S., and N.B.M.S. co-supervised the work. G.D. and N.B.M.S wrote the manuscript with contributions from all co-authors. N.B.M.S. conceived and coordinated the overall project.

## Funding

## Competing interests

The authors declare no competing interests.
