## [Transparent Peer Review file · Nature Communications]

Tunable Octadong and Spindle-Torus Fermi Surfaces in Kramers Nodal Line Metals

Corresponding Author: Dr Niels Schröter

Version 0:

Reviewer comments:

Reviewer #1

(Remarks to the Author)

Please see the attached report.

[Editorial Note: Please see end of file]

Reviewer #2

(Remarks to the Author)

The manuscript by G. Domaine et al. presents ARPES measurements and first-principles calculations of the electronic structure of the 3R polytypes of TaS₂ and NbS₂. The authors report that 3R-TaS₂ and 3R-NbS₂ are both KNLM with octadong and spindle torus Fermi surfaces. The work is interesting and the conclusions are mostly justified by the agreement between experiment and theory. However, already several studies on KNL exist in the literature and it is well known that a noncentrosymmetric achiral material with TRS preserved is a KNLM. The presence of the “almost movable KNL (AMKNL)” is shown in Fig. 2f and Fig. 4d has also been discussed in the literature. Experimentally, KNL has been shown in a few materials already. Additionally, the band structure and the Fermi surface of 3R-NbS₂ is already published in Ref. 16 of this manuscript. The quantization effect in NbS₂ has been published earlier by some of the authors of this work (Ref. 19). So, the novelty aspect of the present work is lacking. There are also several open questions and shortcomings of this work. Considering the aspect of novelty and the shortcomings of the work (see below), acceptance of the manuscript in Nat. Commun. is not recommended.

The presence of the 3R phase in a 2H crystal in more than 50% of the surface area (see Fig. 2a) is quite surprising. It is also not clear whether the appearance of the 3R phase is related to conditions of crystal growth. This is important to present this information because otherwise the work reported may not be reproduced in other crystals studied by other researchers.

Furthermore, the identification of the 3R phase at the surface of a 2H crystal by a band crossing EF in only the GM direction is not convincing. Further studies should be performed to characterize the 3R phase using techniques such as Raman spectroscopy and surface x-ray diffraction.

The authors refer to Ref. 17 and based on that mention that the origin of the 3R phase is related to stacking faults. However, Ref. 17 deals with a different material (SnS)_{1.17}NbS₂ and shows that it has an incommensurate misfit layer structure of two-atom-thick layers of SnS and three-atom-thick sandwiches of NbS₂. Thus, this reference is hardly relevant for 3R-TaS₂.

Additionally, it is not clear why stacking faults would occur to give this particular phase and not the 1T phase, or a new phase, or incommensuration, or a surface reconstruction. LEED may provide a clue, but these data are not presented. The authors should analyze the quantum confinement data rigorously (Note: figure referencing as 5a and 5b is wrong, should be Fig. 3a and 3b) and compare with DFT to determine the possible number of layers of the 3R phase to answer my previous question. Why the quantization effect is not visible in Fig. 2i for the 3R phase? This effect should not be visible in the domains of the 2H phase, is it indeed so? The physical origin of quantum well states for the 3R domains claimed by the authors remains unclear. What is the confining potential between the bulk 2H and the few layers of 3R that presumably grow on it?

The role of soc and the extent of the splitting of the bands caused by soc in the noncentrosymmetric structure is not mentioned. Note that the color scale in Figs. 2f does not have any number. It is not clear whether the authors are able to

resolve the SOC split bands by ARPES and their crossing that forms the KNL. To address this, the authors should provide SOC-split DFT band structures calculated along the same direction as shown in Figs. 2(l-n) to clarify this point. In Figs. 2e and 2j, the authors show the DFT band structure along G_{bar} and M_{bar} . However, it is not properly mentioned whether these are surface calculations. If these are bulk calculations, the authors should specify the k_z values or planes where the calculations were performed. The calculations in Figs. 1c, 1d, and 2f show that 3R-TaS₂ has an AMKNL on the MIMP connecting F and L via an arbitrary k_x - k_z path. This implies that at the surface Brillouin zone, the AMKNL should enforce band crossings along the entire G_{bar} to M_{bar} high-symmetry line. While the authors show these crossings in Figs. 2(l-n), it would be interesting to see ARPES data near M_{bar} to investigate whether these crossings disperse more toward the higher binding energy region and are still enforced.

Additionally, here are a few additional minor comments. In particular, the figure numbering is wrong in several places:

1. In Section B, change the figure number from "Fig. 5a and Fig. 5b" to "Fig. 3a and Fig. 3b."
2. Correct the caption of Fig. 4 to refer to panel 4d (currently labeled as panel e), as the figure contains panels up to d.
3. On page 7, the sentence, "Our DFT calculations of 3R-TaS₂, shown in Fig. 2h and Fig. 2i," should refer to Fig. 2j instead of Fig. 2i.
4. On page 23, change to "Fig. 3a and Fig. 3b" where applicable. In Section C, the sentence, "isoenergy surfaces at different binding energies of 3R-TaS₂ in Fig. 5c," should be corrected to "Fig. 3c."
5. Also, there is no mention of Fig. 2f in the text, the author should clarify this splitting is due to SOC.
6. The caption for Fig. 2f states, "DFT calculation of the band splitting between the two bands crossing the Fermi level on the MIMP." It would be clearer to specify which phase of TaS₂ the authors are referring to.
7. Figures of the crystal structures of 3R and 2H TaS₂ should be provided, as these are not discussed in the paper. The reader has to go through the literature to find what out-of-plane (z) means w.r.t. the primitive lattice vectors (a, b, c).
8. The lattice representation in Fig. 5c is not adequately discussed or referred to in the text. The caption does not fully explain what the figure conveys, and the two axes should be described in the caption.
9. The intensity scale is missing in Fig. 2g.
10. Inconsistency in the high symmetry points b/w Fig. 2c and 2h, one uses bar ($M_{\text{bar}}, k_{\text{bar}}$), other one does not.
11. Correct the typo "ocdong" in the last sentence of page 9.
12. On page 4, the sentence, "obtained by identifying the points $k \sim k + G$, is shown in Fig. 1c," should refer to Fig. 1d.
13. In the tight-binding model, the H1 term uses the notation $(t_1 + t_1^*)_{2 \times 2}$. It should clarify that the suffix reads as $2 \times$ (cross, not alphabet 'x') 2 properly, to avoid confusion.
14. The H7 term in the Hamiltonian contains similar cosine terms. Verify if one of them should be $\cos(k \cdot (-a_1 + a_2 + a_3))$.
15. On page 23, correct "3H-TaS₂" to "3R-TaS₂."

Reviewer #3

(Remarks to the Author)

This study aims to identify a material featuring a highly controllable Kramers nodal line by examining the electronic states of 3R-TaS₂ via both ARPES measurements and first-principles calculations. Kramers nodal lines are expected to exhibit intriguing quantum phenomena such as quantized optical conductivity in thin films or transitions to a different type of topological electronic state (the Kramers Weyl point) under strain. To experimentally observe these phenomena, it is desirable to have high controllability through external fields or carrier doping.

The authors argue that 3R-TaS₂ is an ideal platform offering such high controllability. Specifically, they discuss three main points: (i) 3R-TaS₂ indeed possesses a Kramers nodal line; (ii) when the energy is varied (which can be realized by carrier doping), the Fermi surface evolves from an "Octdong" Fermi surface into a spindle-torus Fermi surface; (iii) the shape (topology) of the line node changes under strain.

Points (i) and (ii) are thoroughly addressed from experimental and theoretical perspectives, and the discussions are clear. The importance of this work is recognized, as it could spur further research into the physical properties of this material. On the other hand, the significance of point (iii) is not fully clear. As described below, if this issue is adequately addressed, I would recommend publication in Nature Communications.

In Sec. II D, the discussion on tuning the line node by strain relies solely on theoretical analysis without any experimental evidence. Including it in the main text might not be necessary, or its importance should be more clearly shown.

It is unclear whether the model Hamiltonian in Eqs. (1)–(8) is correct. At the very least, a comparison with the first-principles results is needed to show how well the effective model approximates the actual material.

The figure references in the text are misaligned and must be corrected.

Reviewer #4

(Remarks to the Author)

Version 1:

Reviewer comments:

Reviewer #1

(Remarks to the Author)

The authors have fully addressed my questions and concerns with extra data and legitimate reasoning. I thus recommend publishing the manuscript in Nature Communications.

Reviewer #2

(Remarks to the Author)

The authors have considerably improved the manuscript and the revised version is more convincing and clearer. However, there remain some important issues listed below that need to be addressed.

1. The inclusion of the XRD data is appreciated but gives rise to a few new questions regarding the XRD of the 3R phase. Why is the 3R (015) peak considerably broader compared to 2H in Fig. 2c? Additionally, while one can identify the (015) XRD peak, the existence of the 104-peak shown at 36.7° and its fitting is not convincing. The peak seems to be narrow and at a slightly smaller 2θ compared to the database, while the fitting shows it to be much broader. This should be clarified. A larger range of data up to $2\theta \sim 85-90^\circ$ should be shown and the 3R (and 2H) peaks identified in this extended range. It will be interesting to estimate the percentage of the 3R phase in the bulk 2H crystal, and an explanation of why it is so large ($\sim 50\%$) at the surface should be provided.
2. I am rather concerned that, as the authors mention, HQ Graphene “did unfortunately not disclose detailed growth conditions”. So, the authors mention nothing about the crystal growth in the Methods. It is likely that the presence of the 3R phase in a 2H crystal, on which their whole study is based, depends on the growth conditions. So, this incomplete information indicates the possibility that, as I mentioned in my earlier report, the amount of the 3R phase in a different batch of crystals from different growers/suppliers might vary significantly. Thus, future researchers, who might want to work further on the present interesting findings reported here, will be put in a disadvantageous position. So, the authors should provide the details of the crystal growth. Alternately, the authors should show the existence of a similar amount of 3R phase at the surface (Fig. 2b) as obtained here on at least two crystals from other sources. This will show that the presence of the 3R phase is independent of any specific condition(s) of crystal growth.
3. The calculated Fermi surface (FS) presented in Fig. 1b does not agree with the FS shown in the 2nd panel of Fig. 4d ($E=EF$), rather it matches with the $E = EF - 50$ meV. The authors should clarify this disagreement. I assume both the FSs are related to 3R TaS₂.
4. From Fig. 3b, it is evident that the AMKNL-induced crossings occur at different k_z values for different k_x , as also noted by the authors: “Note that the gapless nodal line only occurs at specific out-of-plane momenta k_z within the mirror plane, which would technically correspond to a specific photon energy in an ARPES experiment.” However, the DFT cuts shown in Fig. 3c are computed only at $k_z = 0$, where tiny SOC-induced gaps are expected along the GL direction (at $k_y=0$). The caption—“While going from the central hole pocket to the external electron pocket the crossing point transitions from above (red) to below (blue and green) the Fermi level”—may be misleading, as it could suggest that crossings are expected in all k_x cuts at this k_z . Upon close inspection of the newly added green cut in Fig. 3c, a tiny SOC gap can indeed be seen. To avoid confusion, the authors are encouraged to clarify that such a tiny gap is expected at $k_z = 0$, or to provide DFT cuts at the appropriate k_z values where the crossings are truly gapless due to the AMKNL.
5. The authors mention a point about the quantization “Finally, the differences in the band gaps of the 2H and 3R polytypes can naturally lead to the formation of a confinement potential.” However, both 2H and 3R TaS₂ are metals with no band gap, making it confusing which gaps the authors are referring to here.
6. The inclusion of both the no-SOC and SOC band structures in Fig. 3c is appreciated, as it helps readers see the role of SOC. However, this comparison raises an interesting point regarding the splitting of the dog-bone and circular electron pockets around the M and K points, respectively, in the 2H phase. According to the calculations, such pockets arise in the 3R phase due to SOC. However, in the centrosymmetric 2H phase, SOC is not expected to lift the spin degeneracy due to PT symmetry. In Fig. 3c (green cut), the soc split inner band corresponds to the dog-bone pocket, while the outer band gives rise to circular pockets around K. Although this is not directly related to the main objective of the paper, a brief discussion on the possible origin of such similar Fermi pockets in the centrosymmetric 2H phase—resembling the SOC-driven features of the non-centrosymmetric 3R phase—would be helpful for better understanding.

Minor Points:

1. Several typographical errors remain in the manuscript and supplementary material. For example, in the caption of Fig. 9 in the supplement, “polytypes” is misspelled as “opytypes.” Please check for other typos throughout the manuscript.
2. Please specify which Ta core levels are shown in Fig. 9 of the Supplementary Material.
3. In Fig. 3a, the axes values for k_x and k_y are missing and should be added for clarity.
4. Correct the Fig. 3c caption “along three different directions” also in the related text on page 8, now there are 4 cuts.
5. This would be Fig. 3a,c instead of 2a,c: “The Dirac-like dispersion ... touching hole to electron pockets (Fig. 2 a, c) is the predicted hallmark” on page 8.

Reviewer #4

(Remarks to the Author)

I co-reviewed this manuscript with one of the reviewers who provided the listed reports. This is part of the Nature Communications initiative to facilitate training in peer review and to provide appropriate recognition for Early Career

Researchers who co-review manuscripts.

Version 2:

Reviewer comments:

Reviewer #2

(Remarks to the Author)

The report is uploaded as a pdf file "report on NCOMMS-24-82581B with attachment"

[Editorial Note: Please see end of file]

Reviewer #4

(Remarks to the Author)

Author response to comments by Reviewer #1:

Reviewer comment: *In this work by G. Domaine et al., the authors reported tunable band structure with nontrivial topology in the 3R polytype of widely studied transition metal dichalcogenides (TMDCs). Specifically, the authors performed angle-resolved photoemission spectroscopy (ARPES) on TaS₂ and NbS₂. While 3R-NbS₂ crystals are directly available, 3R-TaS₂ phase resides in bulk crystals of 2H-TaS₂ and thus needs to be identified with microARPES spatial mapping.*

Through ARPES measurement, as well as supportive DFT calculation, the authors claimed the observation of “Octdome” Fermi Surface (FS) in 3R-TaS₂ and “Spindle-Torus” FS in 3R-NbS₂, connected by a difference in chemical potential in these two systems. The quality of ARPES data is extraordinary. The theory is reasonably readable to non-expert audiences with some level of knowledge in topology. The manuscript is overall well-written. The observation and conclusion are interesting and insightful, and thus fit the scope of Nature Communications.

Author response: We thank the reviewer for the positive evaluation of our work and for recognizing the interesting and insightful aspects of our study. We appreciate your acknowledgment that the manuscript fits the scope of *Nature Communications*.

Reviewer comment: *However, before I can recommend this manuscript for publication in Nature Communications, there are some questions and suggestions that I hope the authors could address. Please see my comments below.*

1. *From the manuscript in its current form, it's hard to imagine and visualize the different crystal structures and stacking configurations. I strongly recommend the authors include a panel in Fig1 comparing the crystal structures of 1T, 1H, 2H and 3R polytypes of TMDCs*

Author response: We have now included the crystal structure of the 2H and 3R phases in Figure 2. We did not include the crystal structures of every individual polytype since we have now further evidence from XRD that we are indeed looking at the 2H and 3R polytypes.

2. *Where is the natural cleavage plane for the 3R polytype? From the crystal structure, theoretically there could be three different cleavage planes and thus three different surface terminations. The Fermi surfaces measured with ARPES largely depend on the surface termination type. Also, could there be surface reconstructions?*

Author response: The natural cleavage plane of this material is between the TaS₂ layers since the van der Waals forces are significantly weaker than the intralayer covalent bonds. As a result, all the terminations can be expected to be identical since the three layers within the conventional unit cell are simply shifted by translations with respect to each other. Some TaS₂ polytypes, such as 1T-TaS₂ and 2H-TaS₂, are

known to form periodic lattice distortions due to charge density wave formation. However, in the present work we intentionally kept the sample temperature above the known CDW phase transition temperature of the 2H-TaS₂ phase to clearly observe the Kramers nodal line phase. Since we do not detect any backfolded bands from the expected band structure, a surface reconstruction can be excluded at the reported measurement temperature.

3. *The 3R polytype has an out-of-plane lattice constant (c) of ~18 Angstrom. The ARPES measurements presented in this work are performed with UV photons with an energy around 50 eV. This is the most surface-sensitive regime in ARPES measurement, where, according to the universal curve of electron mean-free-path, the probing depth is <4 Angstrom. This suggests that ARPES does not probe the entire unit cell in 3R polytype. Based on this factor, how shall we understand the ARPES spectrum?*

Author response: The effect of the reduced probing depth is so called k_z broadening of the ARPES spectra along the out-of-plane momentum direction k_z [see e.g., Strocov, Journal of Electron Spectroscopy and Related Phenomena 130, 65–78 (2003)]. This effect becomes significant if the unit cell is large compared to the mean free path of the photoelectrons. In our data, this effect is most pronounced in the out of plane dispersion of the valence band of TaS₂ shown in the revised Fig. 4c of the main text due to its strong k_z dispersion.

4. *TaS₂ systems in general host charge density wave (CDW) phases. Though the authors wisely chose a measurement temperature above the CDW long range order phase transition, the system (at least 2H polytype) hosts a short-range CDW at a much higher temperature (PHYSICAL REVIEW B 99,245144(2019)). The measurement temperature is very close to the transition temperature in 2H-TaS₂. In this temperature regime, one would naively expect a strong charge density fluctuation. Does this factor affect either ARPES measurement or DFT calculation?*

Author response: In the work referenced by the reviewer, the effect of the short-range CDW is the suppression of spectral weight at the Fermi level at the K pocket. A similar pseudogap has also been reported in the normal state of 2H-TaSe₂ (Borisenko et al., Phys. Rev. Lett. 100, 196402 (2008)), primarily affecting the K pocket. For these reasons, we do not expect that the CDW transition will significantly impact the band dispersion of the Kramers nodal line along the Γ -M direction, which is the main focus of our work. Furthermore, we deliberately avoided discussing other CDW phases in 3R-TaS₂, as this topic will be covered by us in another upcoming publication that we do not want to cannibalize.

5. *The effect of matrix element on the ARPES spectrum is missing. Could the same features of the Fermi surfaces be observed in ARPES with different incident photon energy and polarization?*

Author response: Unfortunately, we do not have data taken with a different polarization. However, because the neighboring Brillouin zones would have different matrix elements as well, we included in the supplementary a Fermi surface map obtained at 126 eV that includes multiple Brillouin zones. From this data, we see that the only variation is the absolute intensity of the bands, but no new spectral features appear. We have now included also a reference to this supplementary figure in the main text.

FIG. 9. **Fermi surface map of the 3R phase measured with $h\nu = 126$ eV photons.** We can see that even in the neighboring Brillouin zones, where the matrix elements are different, the hole and electron pockets are connected at one point.

6. *The proof of 3R polytype residing in 2H-TaS₂ is a bit handwavy. Stacking faults and misfit in TMDCs could also cause monolayer-like electronic structures (arXiv:2308.02772). The authors attempted to differentiate the single-band FS structure from that of monolayer 1H phase from an electronic perspective. However, the difference among distinct polytypes lies in the crystal structure after all. Thus, a structural probe is the smoking-gun evidence of a such phase mixing. Thus, I suggest additional measurements that reflect the distinct crystal structures in different domains, ideally on the same sample where ARPES was measured. Any of nano X-ray diffraction, spatially resolved Raman scattering and transmission electron microscopy should in principle be sufficient.*

Author response: We have now performed single crystal X-Ray diffraction revealing the simultaneous presence of both the TaS₂ 2H and TaS₂ 3R polytypes in the bulk of our material. We have included this data in the revised Fig. 2(c,d) of the main text. In particular we observe peaks corresponding to unique planes of the 2H and 3R phases in the 2θ dependence in agreement with the reported entries in the Pearson's crystallographic database for these two phases of TaS₂ (Ref. 18 of the revised manuscript). Additionally, the intensity ratio of two peaks (015) and (018) expected for the 3R polytype in the ϕ scan shown in revised Fig. 2d is approximately $3.4:1 \pm 0.12$. This value is once again in agreement with the values reported in the database for the 3R phase of approximately 3.7:1.

Revised Fig. 2 c) Single Crystal X-Ray Diffraction intensity as a function of the scattering angle (2θ) showing peaks belonging to different atomic planes of the 2H (blue) and 3R (green) phases and corresponding in-plane rotation dependence for the (015) and (018) peaks d) exhibiting a ratio of approximately 3.4:1.

Reviewer comment: *I am willing to reconsider this manuscript for publication after the questions above are properly addressed.*

Author response: We hope that we have addressed all of the reviewers concerns and that they can now recommend publication of our manuscript.

Reviewer #2 (Remarks to the Author):

Reviewer comment: *The manuscript by G. Domaine et al. presents ARPES measurements and first-principles calculations of the electronic structure of the 3R polytypes of TaS₂ and NbS₂. The authors report that 3R-TaS₂ and 3R-NbS₂ are both KNLM with octdng and spindle torus Fermi surfaces. The work is interesting and the conclusions are mostly justified by the agreement between experiment and theory.*

Author response: We thank the reviewer for recognizing the interesting aspects of our study and acknowledging the agreement between experiment and theory.

Reviewer comment: *However, already several studies on KNL exist in the literature and it is well known that a noncentrosymmetric achiral material with TRS preserved is a KNLM. The presence of the “almost movable KNL (AMKNL)” is shown in Fig. 2f and Fig. 4d has also been discussed in the literature. Experimentally, KNL has been shown in a few materials already. Additionally, the band structure and the Fermi surface of 3R-NbS₂ is already published in Ref. 16 of this manuscript. The quantization effect in NbS₂ has been published earlier by some of the authors of this work (Ref. 19). So, the novelty aspect of the present work is lacking. There are also several open questions and shortcomings of this work. Considering the aspect of novelty and the shortcomings of the work (see below), acceptance of the manuscript in Nat. Commun. is not recommended.*

Author response: We appreciate the reviewer’s comment and would like to clarify and emphasize two key aspects regarding the novelty and significance of our work.

1) To observe any of the interesting effects predicted for KNL materials—such as the quantization of the optical conductivity—the KNL must cross the Fermi level. This has never been conclusively demonstrated experimentally. There has been a report about a KNL in the charge density wave state in RTe₃ [13, 14], however, no clear splitting or crossing of the nodal line at the Fermi level has been measured. Thus, we believe our work constitutes the first clear experimental evidence for a KNL **metal** where the nodal line is crossing the Fermi level.

2) Furthermore, not all KNL metals behave the same. They can in fact manifest in two different forms—such as the spindle-torus and the Octdong Fermi surface. The Octdong configuration is particularly intriguing because it has been predicted to give rise to a quantized optical conductivity in the thin film limit. However, in previous theoretical work discussing the Octdong Fermi surface [Xie et al., Nat Commun. 12, 3064 (2021)], it was concluded that *“Unfortunately, we have yet to identify realistic materials with pure octdong Fermi surfaces”*. Our work is, to our knowledge, the first to theoretically identify the appearance of an Octdong Fermi surface in a real material, and to experimentally detect it using angle-resolved photoemission spectroscopy (ARPES).

We believe that both of these aspects represent a significant advance in the field. We would also like to note that other reviewers have already recognized the novelty and impact of our work, and we hope these remarks address the concerns raised and clarify the innovative contributions of our study.

Reviewer comment: *The presence of the 3R phase in a 2H crystal in more than 50% of the surface area (see Fig. 2a) is quite surprising. It is also not clear whether the appearance of the 3R phase is related to conditions of crystal growth. This is important to present this*

information because otherwise the work reported may not be reproduced in other crystals studied by other researchers.

Author response: We agree with the reviewer that the presence of the 3R phase at the surface of a cleaved crystal with nominal 2H phase is very surprising. The samples used in our study were purchased from HQ Graphene (Groningen, Netherlands), a widely used supplier in the 2D materials community. According to the supplier, these crystals are characterized using X-ray diffraction (XRD), Raman spectroscopy, and energy-dispersive X-ray spectroscopy (EDX) and are sold as 2H-TaS₂ (see <https://www.hqgraphene.com/TaS2.php>). Upon request, the supplier did unfortunately not disclose detailed growth conditions beyond stating that the samples were synthesized via chemical vapor transport.

The Raman modes between the 3R- and 2H- phases of the transition metal sulphides are extremely similar (see e.g. X. Zhou *et al.*, doi:[10.48550/arXiv.2502.11977](https://doi.org/10.48550/arXiv.2502.11977), and W. G. McMullan, J. C. Irwin, *Solid State Communications*. 45, 557–560 (1983)) and EDX can be expected to be similar as well. However, our own XRD characterization detects the presence of a minority phase of 3R-TaS₂ in the bulk (see answer to the next question). It is therefore likely that XRD characterization is not performed on every batch at HQ graphene. The exact origin of this minority phase is not clear, but literature reports suggest that an excess of tantalum can lead to the preferential growth of 3R-TaS₂ from which single crystals can be isolated (see Y. Gotoh, J. Akimoto, Y. Oosawa, *Journal of Alloys and Compounds*. 270, 115–118 (1998)).

Another possible mechanism for the formation of the 3R phase is the presence of partial dislocations, as recently discussed in MoS₂ by X. Zhou *et al.* (see doi:[10.48550/arXiv.2502.11977](https://doi.org/10.48550/arXiv.2502.11977)). These authors observed both 2H and 3R phases in mechanically exfoliated flakes from 2H-MoS₂ and speculated that the exfoliation process itself could induce a phase transition between energetically equivalent stacking configurations. It is also conceivable that crystal cleavage occurs preferentially near dislocations related to the formation of the 3R phase, making it more prevalent near the surface.

In conclusion, while we cannot definitively determine the origin of the 3R-TaS₂ phase in our samples, similar observations of phase coexistence in cleaved 2H-TMDC crystals have been reported. Importantly, our main conclusion—the discovery of Octdong and spindle-torus Fermi surfaces in 3R-TaS₂ and 3R-NbS₂—remains unaffected by the formation mechanism of these phases.

Reviewer comment: *Furthermore, the identification of the 3R phase at the surface of a 2H crystal by a band crossing EF in only the GM direction is not convincing. Further studies should be performed to characterize the 3R phase using techniques such as Raman spectroscopy and surface x-ray diffraction.*

Author response: We thank the reviewer for this helpful suggestion. Whilst Raman measurements cannot distinguish these two phases conclusively because the spectra are extremely similar (see e.g. X. Zhou *et al.*, doi:[10.48550/arXiv.2502.11977](https://doi.org/10.48550/arXiv.2502.11977), and W. G. McMullan, J. C. Irwin, *Solid State Communications*. 45, 557–560 (1983)), we have now

carried out single crystal x-ray diffraction (SC-XRD) experiments which unambiguously reveal the simultaneous presence of both the 2H and 3R polytypes. We have included this data in the revised Fig. 2(c,d) of the main text. Using a four-circle and six-axis single crystal diffractometer, several Bragg-reflections uniquely related to either the 2H or the 3R polytype could be identified. Their position in k-space and their intensity (ratios) match with the entries in the Pearson's database (new Ref. 18). For instance, two indicative reflections for the 3R polytype [(015) and (018)] only exhibit an intensity ratio of approximately $3.4:1.0 \pm 0.12$, as shown more clearly in the transverse scan probing the integrated intensity curve in Fig. 2d, which is in agreement with the values reported in the database for the 3R phase of approximately 3.7:1.0.

Revised Fig. 2 c) Single Crystal X-Ray Diffraction intensity as a function of the scattering angle (2θ) showing peaks belonging to different atomic planes of the 2H (blue) and 3R (green) phases and corresponding in-plane rotation dependence for the (015) and (018) peaks **d)** exhibiting a ratio of approximately 3.4:1.

Furthermore, the information obtained from ARPES points to the identification of the new phase as the 3R polytype. The two fundamental building blocks of the TaS₂ TMDCs are the 1T and 1H monosheets, in whose center the Ta atom is located either in an octahedral or in a trigonal prismatic coordination, respectively. These two Ta coordinations differ from each other by significant differences in the energy of the Ta core levels and in the band structure (Wang, Y.D., Yao, W.L., Xin, Z.M. et al., Nat Commun 11, 4215, 2020, H. P. Hughes and J. A. Scarfe, Journal of Physics: Condensed Matter 8, 1996, J. A. Scarfe and H. P. Hughes, Journal of Physics: Condensed Matter 1, 1989). From the results of the band structure mapping and the core level XPS (see Fig. 8 of the supplementary materials reproduced below) as well as from the XRD data, we can exclude all polytypes involving Ta in an octahedral coordination. This leaves us with only the 2H (a, b, and c), 3R, and 4H (a and c) polytypes. Of these, only

the 3R polytype has a single Ta site in the primitive unit cell, leading to a single band at the Fermi level.

FIG. 8. **Tantalum core levels.** The core levels of a 2H patch and a 3R patch showing almost identical features, suggesting the presence of a single type of coordination of the Ta atoms. Notice that the core levels for the 1T polytype, as well as other polytypes including 1T monolayers exhibit a different number of core levels peaks with different relative intensities [36, 37].

Reviewer comment: *The authors refer to Ref. 17 and based on that mention that the origin of the 3R phase is related to stacking faults. However, Ref. 17 deals with a different material (SnS)_{1.17}NbS₂ and shows that it has an incommensurate misfit layer structure of two-atom-thick layers of SnS and three-atom-thick sandwiches of NbS₂. Thus, this reference is hardly relevant for 3R-TaS₂. Additionally, it is not clear why stacking faults would occur to give this particular phase and not the 1T phase, or a new phase, or incommensuration, or a surface reconstruction. LEED may provide a clue, but these data are not presented.*

Author response: We acknowledge and apologize for the inappropriate reference. We replaced the original reference with two new references related to similar sulphur-based TMDCs [J. Strachan, A. F. Masters, T. Maschmeyer, ACS Appl. Energy Mater. 4, 7405–7418 (2021); Y. Yang et al., American Mineralogist. 107, 997–1006 (2022)]. In particular the first reference mentions that the growth of rhombohedral phase TMDCs has been associated with the presence of screw-dislocations, which results in a continuous layer forming a spiral in the 3R-phase, which cannot result in the 2H-phase.

In the 1T phase mentioned by the reviewer, the transition metal has an octahedral coordination, while in the 2H phase it has a trigonal prismatic coordination, the same as in the 3R phase. This means that the 1T phase cannot appear as a result of stacking faults in a 2H crystal, while the 3R can be realized by changing the type of stacking. Moreover, the 1T phase has been widely studied and as such can be directly excluded based on a direct

comparison of the band structures and its core levels energies (Wang, Y.D., Yao, W.L., Xin, Z.M. et al., Nat Commun 11, 4215 2020, H. P. Hughes and J. A. Scarfe, Journal of Physics: Condensed Matter 8, 1996 and J. A. Scarfe and H. P. Hughes, Journal of Physics: Condensed Matter 1, 1989). See also Fig. 8 of the supplementary materials shown below.

The fact that only two fundamental building blocks exist (the octahedral and trigonal prismatic one) also constrains the number of possible phases of TaS₂ which are already known. Even by assuming the appearance of a previously unobserved polytype, we know from the binding energy of the Ta core levels that the new phase is composed of trigonal prismatic building blocks only. This, combined with the number of bands at the Fermi level, which originate from the transition metals d-orbitals, allows us to exclude the polytypes with more than one Ta site in their primitive unit cell. This is due to a well-known result of band theory, which states that for every atom contributing partially filled orbitals to the fermi level, there will be a single (spin-degenerate) band. Taking all these experimental facts together, we are thus left with only the 3R-polytype as a possible option.

FIG. 8. Tantalum core levels. The core levels of a 2H patch and a 3R patch showing almost identical features, suggesting the presence of a single type of coordination of the Ta atoms. Notice that the core levels for the 1T polytype, as well as other polytypes including 1T monolayers exhibit a different number of core levels peaks with different relative intensities [36, 37].

Reviewer comment: *The authors should analyze the quantum confinement data rigorously (Note: figure referencing as 5a and 5b is wrong, should be Fig. 3a and 3b) and compare with DFT to determine the possible number of layers of the 3R phase to answer my previous question.*

Author response: We thank the reviewer for this suggestion. We have now included in the revised Fig. 4(a,b) of the main text the results from the slab calculations corresponding to 5-TaS2 layers, showing agreement with the experimental data. We also corrected the issues with the referencing.

Revised Fig. 4 a) band structure of 3R-TaS2 at 72 eV on the mirror plane, showing a quantization of the valence band and **b)** the corresponding 5-layers slab calculation obtained from DFT.

Reviewer comment: *Why the quantization effect is not visible in Fig. 2i for the 3R phase?*

Author response: If the k_z dispersion is weak, as expected for the conduction band of the 3R phase from DFT, the quantum well states are very closely spaced in energy. This is why the size confinement is not clearly visible in our data in Fig. 2i. In contrast, the k_z dispersion of the valence band is much stronger, which means that quantum size effects would lead to a larger energy splitting of quantum well states.

Reviewer comment: *This effect should not be visible in the domains of the 2H phase, is it indeed so?*

Author response: Interestingly, the quantized levels can also be observed in some parts of the sample where we have only a few layers of the 2H phase, and we have included a new Fig 7 in the supplementary material showing these spectra that we mention in the revised version of the main text.

FIG. 7. Quantization of the valence band in the 2H phase. Just as in the 3R phase, certain areas of the 2H phase exhibit a quantization of the valence band.

Reviewer comment: *The physical origin of quantum well states for the 3R domains claimed by the authors remains unclear. What is the confining potential between the bulk 2H and the few layers of 3R that presumably grow on it?*

Author response: We thank the reviewer for this interesting question. In our revised manuscript, we discuss three possible origins for the confining potential of the quantum well states observed in TaS₂.

- 1) In the case of upwards surface band bending of the valence band (which would be consistent with the slight hole doping of the band structure compared to the charge neutrality level expected from DFT), there will be a confining potential due to the valence band edge moving to larger binding energies at higher depths away from the surface. It has been suggested that this is the origin of the quantum well states observed in the valence band of TaSe₂ in Ref. 20 [Chinese Physics B 30, 047305 (2021)].
- 2) Since the band structure of the 2H- and 3R- phases are different with slightly different band gaps, it is natural to assume that there can be a confining potential. In fact, in conventional III-V semiconductors, a confining potential at such a stacking fault leading to phase boundaries is well known to form quantum well states that can be exploited for optoelectronic applications, see e.g. Akopian, N., et al., Nano letters 10.4 (2010): 1198-1201.
- 3) A third possibility is that the quantized states observed in the valence band are in fact surface states, in which case the confining potential is due to the change of crystal structure between surface and bulk. We mention this possibility in the main text.

Reviewer comment: *The role of soc and the extent of the splitting of the bands caused by soc in the noncentrosymmetric structure is not mentioned. Note that the color scale in Figs. 2f does not have any number. It is not clear whether the authors are able to resolve the soc*

split bands by ARPES and their crossing that forms the KNL. To address this, the authors should provide SOC-split DFT band structures calculated along the same direction as shown in Figs. 2(l-n) to clarify this point.

Author response: We thank the reviewer for this helpful suggestion. We have now added a number to the revised figure that the referee was referring to. We have furthermore added a new figure (Fig. 3 c) in the revised main text (see below) where we show the DFT band structure with and without spin-orbit coupling and compare it to the measured bands, which clearly shows that we are able to resolve the spin-split bands by ARPES. We have included a comment to the main text highlighting our ability to resolve this splitting.

FIG. 3. Octadong Fermi surface formed by the nodal line piercing E_F in 3R-TaS₂. a) ARPES and DFT ($k_z = 0$) Fermi surfaces showing the octadong formed by the hole and electron pockets. b) Calculated splitting between the two bands crossing the Fermi level on the mirror plane, showing the presence of a nodal line winding twice around the mirror plane. Panels c) show the dispersion along three different directions perpendicular to the mirror plane on both sides of the octadong touching point along the cuts in a), both for the ARPES data and the DFT calculations ($k_z = 0$). While going from the central hole pocket to the external electron pocket the crossing point transitions from above (red) to below (blue and green) the Fermi level, as predicted for the octadong Fermi surface in Ref. [6], thus crossing the Fermi level (purple). The dotted lines represent the degenerate bands in the absence of SOC.

Reviewer comment: *In Figs. 2e and 2j, the authors show the DFT band structure along G_{bar} and M_{bar} . However, it is not properly mentioned whether these are surface calculations. If these are bulk calculations, the authors should specify the k_z values or planes where the calculations were performed.*

Author response: We have now edited the figure to accommodate the information that was missing. Indeed, the DFT results were bulk calculations, and we have now also included the results for different k_z planes.

Reviewer comment: *The calculations in Figs. 1c, 1d, and 2f show that 3R-TaS₂ has an AMKNL on the MIMP connecting F and L via an arbitrary k_x - k_z path. This implies that at the surface Brillouin zone, the AMKNL should enforce band crossings along the entire G_{bar} to M_{bar} high-symmetry line. While the authors show these crossings in Figs. 2(l-n), it would be interesting to see ARPES data near M_{bar} to investigate whether these crossings disperse more toward the higher binding energy region and are still enforced.*

Author response: We have now included in the figure in the revised Fig. 3c showing an additional cut closer to the M point, indicating that the crossing indeed disperses downwards and is still enforced, in agreement with the DFT results.

FIG. 3. Octadong Fermi surface formed by the nodal line piercing E_F in 3R-TaS₂. a) ARPES and DFT ($k_z = 0$) Fermi surfaces showing the octadong formed by the hole and electron pockets. b) Calculated splitting between the two bands crossing the Fermi level on the mirror plane, showing the presence of a nodal line winding twice around the mirror plane. Panels c) show the dispersion along three different directions perpendicular to the mirror plane on both sides of the octadong touching point along the cuts in a), both for the ARPES data and the DFT calculations ($k_z = 0$). While going from the central hole pocket to the external electron pocket the crossing point transitions from above (red) to below (blue and green) the Fermi level, as predicted for the octadong Fermi surface in Ref. [6], thus crossing the Fermi level (purple). The dotted lines represent the degenerate bands in the absence of SOC.

Reviewer comment: *Additionally, here are a few additional minor comments. In particular, the figure numbering is wrong in several places:*

1. In Section B, change the figure number from “Fig. 5a and Fig. 5b” to “Fig. 3a and Fig. 3b.”
2. Correct the caption of Fig. 4 to refer to panel 4d (currently labeled as panel e), as the figure contains panels up to d.
3. On page 7, the sentence, “Our DFT calculations of 3R-TaS₂, shown in Fig. 2h and Fig. 2i,” should refer to Fig. 2j instead of Fig. 2i.
4. On page 23, change to “Fig. 3a and Fig. 3b” where applicable. In Section C, the sentence, “isoenergy surfaces at different binding energies of 3R-TaS₂ in Fig. 5c,” should be corrected to “Fig. 3c.”
5. Also, there is no mention of Fig. 2f in the text, the author should clarify this splitting is due

to SOC.

6. The caption for Fig. 2f states, "DFT calculation of the band splitting between the two bands crossing the Fermi level on the MIMP." It would be clearer to specify which phase of TaS₂ the authors are referring to.
7. Figures of the crystal structures of 3R and 2H TaS₂ should be provided, as these are not discussed in the paper. The reader has to go through the literature to find what out-of-plane (z) means w.r.t. the primitive lattice vectors (a,b,c).
8. The lattice representation in Fig. 5c is not adequately discussed or referred to in the text. The caption does not fully explain what the figure conveys, and the two axes should be described in the caption.
9. The intensity scale is missing in Fig. 2g.
10. Inconsistency in the high symmetry points b/w Fig. 2c and 2h, one uses bar ($M_{\bar{}}$, $k_{\bar{}}$), other one does not.
11. Correct the typo "ocdong" in the last sentence of page 9.
12. On page 4, the sentence, "obtained by identifying the points $k \sim k + G$, is shown in Fig. 1c," should refer to Fig. 1d.
13. In the tight-binding model, the H1 term uses the notation $(t_1 + t_1^*)^2$. It should clarify that the suffix reads as 2 x (cross, not alphabet 'x') 2 properly, to avoid confusion.
14. The H7 term in the Hamiltonian contains similar cosine terms. Verify if one of them should be $\cos(k \cdot (-a_1 + a_2 + a_3))$.
15. On page 23, correct "3H-TaS₂" to "3R-TaS₂."

Author response: We have now fixed all of the labelling mistakes and typos and added the crystal structures as well as a clarification that the splitting on the mirror plane is due to SOC.

Reviewer #3 (Remarks to the Author):

Reviewer comment: *This study aims to identify a material featuring a highly controllable Kramers nodal line by examining the electronic states of 3R-TaS₂ via both ARPES measurements and first-principles calculations. Kramers nodal lines are expected to exhibit intriguing quantum phenomena such as quantized optical conductivity in thin films or transitions to a different type of topological electronic state (the Kramers Weyl point) under strain. To experimentally observe these phenomena, it is desirable to have high controllability through external fields or carrier doping. The authors argue that 3R-TaS₂ is an ideal platform offering such high controllability. Specifically, they discuss three main points: (i) 3R-TaS₂ indeed possesses a Kramers nodal line; (ii) when the energy is varied (which can be realized by carrier doping), the Fermi surface evolves from an "Octdong" Fermi surface into a spindle-torus Fermi surface; (iii) the shape (topology) of the line node changes under strain. Points (i) and (ii) are thoroughly addressed from experimental and theoretical perspectives, and the discussions are clear. The importance of this work is recognized, as it could spur further research into the physical properties of this material.*

Author response: We thank the reviewer for stressing the importance of our work and the fact that the vast majority of our claims are thoroughly supported by experiments and theory.

Reviewer comment: *On the other hand, the significance of point (iii) is not fully clear. As described below, if this issue is adequately addressed, I would recommend publication in Nature Communications.*

In Sec. II D, the discussion on tuning the line node by strain relies solely on theoretical analysis without any experimental evidence. Including it in the main text might not be necessary, or its importance should be more clearly shown.

Author response: As the referee correctly points out, here we only propose that such a transition should be possible in principle, with an experimental verification left for future work. However, we think that besides the identification of a new platform to investigate Kramers nodal line metals, it would also be appropriate to make proposals for work that we and other researchers can work on in the future. Since such a transition would potentially significantly change the physical properties of Kramers nodal line metals, we think that it is important enough to remain in the main text.

Reviewer comment: *It is unclear whether the model Hamiltonian in Eqs. (1)–(8) is correct. At the very least, a comparison with the first-principles results is needed to show how well the effective model approximates the actual material.*

Author response: In the previous version of our manuscript, our model was only intended as the generic minimal tight-binding model consistent with the crystal symmetries. We thank the referee for their suggestion to make the relation between DFT and the tight-binding model clearer to highlight the relevance of the model. To achieve a reasonable agreement with the DFT, we have extended our tight-binding model to 17 hopping parameters, which include up to 5th NN terms. We provide in Fig. 6c,d a comparison of our fit and the DFT bands along different cuts. Deviations are expected in a two-band model due to the hybridization of neighboring bands seen in the DFT that cannot be captured by the minimal two-band model. Nevertheless, the connectivity of nodal lines as seen in DFT is qualitatively reproduced in our revised model. The conclusion that a change in the out-of-plane hopping strength determines the connectivity of nodal lines also holds for our new, more realistic model.

FIG. 6. Tunability of nodal line connectivity with strain. a-b) Splitting between the two bands crossing the Fermi level showing two different types of connectivity. In a) the parameters of the tight-binding model are the ones obtained from fitting the model to the bands computed by DFT. On the other hand in b), to capture the effect of a lattice deformation, the out-of-plane hopping amplitudes are multiplied by a factor of 1/10, see Methods Sec. IV C for details. The figures also show the mapping of the nodal lines onto the mirror plane torus (respectively viewed from an elevation of 30° and -30°), where it is apparent that the configuration with the reduced out-of-plane hopping does not wind around the torus, and as such is in general not guaranteed to cross the Fermi level. On the other hand, the configuration with a stronger out-of-plane hopping fully winds twice around the torus, and is thus guaranteed to pierce the Fermi surface. Notice also that the nodal line winding along the meridian on the right-hand side of the torus corresponds to the KNL pinned along the high-symmetry path $\Gamma - T$.

Reviewer comment: *The figure references in the text are misaligned and must be corrected.*

Author response: We corrected all the labeling of the figures.

Reviewer #4 (Remarks to the Author):

Reviewer comment: *I co-reviewed this manuscript with one of the reviewers who provided the listed reports. This is part of the Nature Communications initiative to facilitate training in peer review and to provide appropriate recognition for Early Career Researchers who co-review manuscripts.*

Response to the comments by the reviewers, round 2

Author response to comments by Reviewer #1:

Reviewer comment:

The authors have fully addressed my questions and concerns with extra data and legitimate reasoning. I thus recommend publishing the manuscript in Nature Communications.

Author response: We thank the reviewer for recommending our work for publication.

Author response to comments by Reviewer #2:

Reviewer comment:

The authors have considerably improved the manuscript and the revised version is more convincing and clearer. However, there remain some important issues listed below that need to be addressed.

Author response: We thank the reviewer for recognizing the considerable improvements made to make the current version more convincing and clear.

Reviewer comment:

1. The inclusion of the XRD data is appreciated but gives rise to a few new questions regarding the XRD of the 3R phase. Why is the 3R (015) peak considerably broader compared to 2H in Fig. 2c? Additionally, while one can identify the (015) XRD peak, the existence of the 104-peak shown at 36.7° and its fitting is not convincing. The peak seems to be narrow and at a slightly smaller 2θ compared to the database, while the fitting shows it to be much broader. This should be clarified. A larger range of data up to $2\theta \sim 85-90^\circ$ should be shown and the 3R (and 2H) peaks identified in this extended range. It will be interesting to estimate the percentage of the 3R phase in the bulk 2H crystal, and an explanation of why it is so large ($\sim 50\%$) at the surface should be provided.

Author response: The broadening of the peaks in XRD is inversely proportional to the thickness of the corresponding phase. The significantly broader peaks for the 3R phase are consistent with the identification of the 3R phase as a few-layers thick domains, as also suggested by the quantization effects that we report in the ARPES data. We have now also included in the Supplementary material a section where we analyze the width of the XRD peaks, from which we estimate the thickness of the phase averaged over the full bulk volume to be approximately 6 monolayers. This number is in good agreement

with our comparison of our quantum well state simulation to the ARPES data in Fig. 4a-b, which suggests a thickness of around five monolayers. We have also included a reference to this section of the Supplementary material in the revised main text.

momentum, implying that it could directly exhibit a quantization of the optical conductivity. Based on the number of quantized bands, we estimate that the 3R phase should be approximately 5-monolayers thick. This value is also supported by the analysis of the width of the fitted XRD peaks (see the Supplementary material section VII G).

And the revised supplementary materials contain the following section:

G. XRD peak analysis

For the two fitted peaks for the 3R phase in Fig. 2c we obtain a Full-Width at Half Maximum (FWHM) of approximately 0.04 radians. Using the Scherrer equation we estimate an approximate thickness of 37 Å. Given a lattice parameter $c \approx 18$ Å, we get approximately two unit cells, which correspond to 6 1H monolayers, in good agreement with the 5 quantized bands that we see from ARPES and report in Fig. 4.

Furthermore, the data base refers to bulk-like samples, while our study deals with ultra-thin inclusions of the R3 phase so that the 2Theta positions might differ. More importantly the precise quantification of the very weak (104) reflection in this experimental setup using a Theta-2Theta scan (Bruker D8 discover) is difficult due to the fact that it is almost hidden by the background and the tail of the much stronger 2H (103) reflection. To address the Referee's concern, we have carried out a full single crystal structure analysis of the two phases which not only provides clear evidence that our samples consist of the 2H and the 3R phase but which also allowed to derive their relative abundance within the thickness probed by the Ga-K-alpha x-rays [$1/(\mu)=16\mu\text{m}$]. Here we collected in a six-circle geometry 40-50 reflections per phase, which were subsequently averaged to 19 symmetry independent ones using point group symmetry 3m and 6mm for the 3R and the 2H phase, respectively. The results of this refinement procedure are described in more detail in the revised supplementary materials section G.

From this refinement we can estimate the bulk volume ratio between the 2H- and the 3R-phase in the sample is equal to approximately 8. This volume ratio deviates from the surface ratio of the ARPES results where the 3R-phase appears in almost equal magnitude to the 2H phase. We speculate that the enhanced appearance of the 3R phase in the surface sensitive ARPES spectra is due to the fact that there can be a preferential cleaving site in the vicinity of the stacking fault between the 2H and 3R site, which could lead to an increased appearance of the 3R phase near the surfaces of our

cleaved crystals. An alternative hypothesis has been recently suggested that the cleaving process itself could lead to additional stacking faults leading to the 3R phase in MoS₂ [see Zhou et al., arXiv:2502.11977].

We have added the estimation of the bulk volume fraction of the 3R-phase in the main text and comment on the deviation of the surface fraction from the bulk results:

of the fitted XRD peaks (see the Supplementary material section VII G). To get a reliable estimate of the bulk 2H:3R volume ratio in the crystal, we performed additional quantitative single-crystal X-ray diffraction experiments which allowed us to estimate a bulk volume ratio of approximately 8:1 in favor of the 2H phase (see the Supplementary material section VII G). This partially deviates from the ARPES data, which show nearly equal surface contributions of 2H and 3R. We attribute the enhanced presence of the 3R phase at the surface either to preferential cleaving at 2H and 3R stacking faults or to cleaving-induced stacking faults [21].

Reviewer comment:

2. I am rather concerned that, as the authors mention, HQ Graphene "did unfortunately not disclose detailed growth conditions". So, the authors mention nothing about the crystal growth in the Methods. It is likely that the presence of the 3R phase in a 2H crystal, on which their whole study is based, depends on the growth conditions. So, this incomplete information indicates the possibility that, as I mentioned in my earlier report, the amount of the 3R phase in a different batch of crystals from different growers/suppliers might vary significantly. Thus, future researchers, who might want to work further on the present interesting findings reported here, will be put in a disadvantageous position. So, the authors should provide the details of the crystal growth. Alternately, the authors should show the existence of a similar amount of 3R phase at the surface (Fig. 2b) as obtained here on at least two crystals from other sources. This will show that the presence of the 3R phase is independent of any specific condition(s) of crystal growth.

Author response:

We thank the reviewer for their thoughtful comments. We fully agree that detailed information about the crystal growth process and conditions would be valuable for understanding the origin of the observed 3R-TaS₂ phase. However, as we noted, HQ Graphene does not disclose specific growth conditions—an unfortunate but common

limitation when working with commercially sourced samples, which are now widely used for experimental studies in the condensed matter community.

In order to provide the reader with more information about the potential origin of the 3R-TaS₂ phase found in our samples, we have performed additional quantitative single crystal x-ray diffraction experiments, which allowed us to construct a detailed model of the atomic structure of the two phases, as described in detail in supplementary material section G. From this we have found that not all of the Tantalum atoms appear at the correct position, but a small percentage of around 10(5)% are going into an interstitial site above and below the octahedrally coordinated main Ta site. Occupation of the same interstitial sites has previously been reported for a 3R-TaS₂ sample that was grown with excess of Ta [Y. Gotoh et al., *J. Alloys Compd.* 270, 115-118 (1998)]. In fact, there have been many reports for transition metal dichalcogenides where the 3R-phase is formed in the presence of self-intercalation of the transition metal into interstitial sites, which has been speculated to be because the 3R-phase is energetically better suited to host interstitial atoms than polytypes with other stacking configurations [see e.g. H. Wang et al., *Nat Commun.* 15, 2541 (2024)].

In this respect our sample and that discussed in the paper of Gotoh are quite close and differ only in the presence of the second Ta site, which, however, is only 10% of the total concentration, which is almost within the experimental uncertainty. This allows to conclude that - despite the fact that the preparation conditions might differ in some detail from study to study - the resulting real structure in the samples are very well in agreement to each other.

One route for potential reproduction studies is therefore to use samples with a small excess of Ta that goes into interstitial sites as is already described in the literature such as the one mentioned above. An alternative route is to receive samples from HQ graphene, which remain commercially available. In fact, we would like to emphasize that our observations of mixed 2H/3R domains are consistent across multiple crystals from the same supplier. We have cleaved more than 7 samples in total (some recleaves of the same samples). In the revised supplementary materials (Fig. 11, also shown below), we show a representative result obtained from another sample.

We have added a paragraph to the revised main text to suggest strategies for potential replication strategies:

“Our detailed single-crystal X-ray diffraction experiments revealed a small amount of disorder in our samples where about 10(5)% Ta atoms from the main octahedral site are redistributed into interstitial positions. Such self-intercalation effects have been reported to favour the 3R-polytype in other TMDCs such as NbS₂ [H. Wang et al., *Nat Commun.* 15, 2541 (2024)], and samples with excess Ta have been shown to form 3R-

Ta_{1.08}S₂ with Ta occupying the same interstitial positions as found in our study [Y. Gotoh et al., J. Alloys Compd. 270, 115-118 (1998)]. Possible replication strategies may therefore have to source samples from the same supplier or to develop alternative growth recipes that favors the occupation of interstitial sites whilst maintaining stoichiometry, or alternatively study TaS₂ samples that were grown with an excess of Ta to form the 3R-polytype.”.

FIG. 11. Nodal line in another sample Same analysis as in Fig. 3 but on a 3R domain on a different sample.

The reviewer suggests measuring at least two more crystals from other sources as a way to show that the 3R phase is independent of growth conditions. However, we respectfully disagree that this would be a helpful for our readers, unless accompanied by a systematic study of growth parameters and their impact on the phase diagram of TaS₂—which is well beyond the scope of our current work. Measuring a small number of samples from different suppliers would only introduce additional uncontrolled growth variables, which are unlikely to be helpful to readers who would like to reproduce our results.

We also note that the focus of our study is not on the synthesis or stability of the 3R-TaS₂ phase itself, but rather on the electronic properties of this phase—specifically, the discovery of a Kramers nodal line metal with an Octadong-Fermi surface, the first time that such a Fermi surface has been observed in any material. Our results will likely motivate future synthesis-focused investigations, but we believe that including arbitrary additional samples from other suppliers would not substantively strengthen our claims, and may in fact obscure the key message of the paper.

Reviewer comment:

3. The calculated Fermi surface (FS) presented in Fig. 1b does not agree with the FS shown in the 2nd panel of Fig. 4d ($E=EF$), rather it matches with the $E = EF-50$ meV. The authors should clarify this disagreement. I assume both the FSs are related to 3R-TaS₂.

Author response: We thank the referee for pointing out this apparent inconsistency. Fig. 1b was mostly intended as a generic Fermi Surface that one can have for a 3R phase in TMDCs for a special filling in order to highlight how the nodal lines might arise and why one might expect different types of Fermi surfaces, rather than presenting the Fermi surface of 3R-TaS₂. We have now clarified this in the caption of Fig. 1.

Reviewer comment:

4. From Fig. 3b, it is evident that the AMKNL-induced crossings occur at different k_z values for different k_x , as also noted by the authors: "Note that the gapless nodal line only occurs at specific out-of-plane momenta k_z within the mirror plane, which would technically correspond to a specific photon energy in an ARPES experiment." However, the DFT cuts shown in Fig. 3c are computed only at $k_z=0$, where tiny SOC-induced gaps are expected along the GL direction (at $k_y=0$). The caption—"While going from the central hole pocket to the external electron pocket the crossing point transitions from above (red) to below (blue and green) the Fermi level"—may be misleading, as it could suggest that crossings are expected in all k_x cuts at this k_z . Upon close inspection of the newly added green cut in Fig. 3c, a tiny SOC gap can indeed be seen. To avoid confusion, the authors are encouraged to clarify that such a tiny gap is expected at $k_z=0$, or to provide DFT cuts at the appropriate k_z values where the crossings are truly gapless due to the AMKNL.

Author response: We thank the reviewer for pointing out this potential source of misunderstanding. What is meant with that statement is that as the bands disperse downwards for any k_z in the mirror plane. Therefore, the crossing point is also forced to transition from above to below the Fermi level. We have clarified this in the caption of Fig. 4 in the manuscript. We have also clarified the origin of the small splitting seen in the last panel.

Reviewer comment:

5. The authors mention a point about the quantization "Finally, the differences in the band gaps of the 2H and 3R polytypes can naturally lead to the formation of a confinement potential." However, both 2H and 3R TaS₂ are metals with no band gap, making it confusing which gaps the authors are referring to here.

Author response: We thank the referee for pointing out this potential source of confusion. We have inserted this sentence to answer a referees question about the origin of the confining potential leading to quantum well states. It is well known that stacking semiconductors with different band gaps can realize such a confining potential, and similarly at metal interfaces, relative and hybridization gaps can also lead to a confining potential, see for instance [T.-C. Chiang, Surface Science Reports 39, 181–235 (2000)]. We have now included this reference in the revised main text, see below:

would not have to be quantized. Finally, beyond the well-known quantum-well confinement at III–V semiconductor phase boundaries —exploited in optoelectronic devices [23]— similar confinement potentials are also known to arise in metallic quantum wells because of relative and hybridization gaps [24].

Reviewer comment:

6. The inclusion of both the no-SOC and SOC band structures in Fig. 3c is appreciated, as it helps readers see the role of SOC. However, this comparison raises an interesting point regarding the splitting of the dog-bone and circular electron pockets around the M and K points, respectively, in the 2H phase. According to the calculations, such pockets arise in the 3R phase due to SOC. However, in the centrosymmetric 2H phase, SOC is not expected to lift the spin degeneracy due to PT symmetry. In Fig. 3c (green cut), the soc split inner band corresponds to the dog-bone pocket, while the outer band gives rise to circular pockets around K. Although this is not directly related to the main objective of the paper, a brief discussion on the possible origin of such similar Fermi pockets in the

centrosymmetric 2H phase—resembling the SOC-driven features of the non-centrosymmetric 3R phase—would be helpful for better understanding.

Author response:

We thank the referee for raising this interesting point. The symmetry reduction from SG 194 to 160 may generally result in drastic changes in the band structure because the symmetry group reduces from 24 to 6 group elements. So indeed the similarity of Fermi surfaces may be surprising. Yet, the TaS₂ planes are only slightly perturbed in the in-plane direction for a different stacking. Further, since the bands at the Fermi energy are comparably well separated from other bands, a hybridization due to symmetry reduction starting from 2H will likely involve mostly the two similarly dispersing bands at E_F. When the symmetry in 2H that protects the Dirac line along A-L is broken, the two formerly PT degenerate bands are pushed apart. The band that stays at E_F will remain well-separated and will not react much to the deformation of other bands. Thus, the non-spin split shape of the upper band's Fermi pocket can be expected to stay similar. When SOC is added, the spin-split Fermi surfaces follow the inherited Fermi surfaces, making them more likely to exhibit Fermi surfaces in the 3R-phase similar to those of the 2H-phase. Whilst we appreciate this discussion, we also agree with the referee that it is not directly related to the main objective of the paper and we have therefore decided not to include it into the already relatively long main text.

Reviewer comment:

Minor Points:

1. Several typographical errors remain in the manuscript and supplementary material. For example, in the caption of Fig. 9 in the supplement, "polytypes" is misspelled as "opytypes." Please check for other typos throughout the manuscript.
2. Please specify which Ta core levels are shown in Fig. 9 of the Supplementary Material.
3. In Fig. 3a, the axes values for k_x and k_y are missing and should be added for clarity.
4. Correct the Fig. 3c caption "along three different directions" also in the related text on page 8, now there are 4 cuts.
5. This would be Fig. 3a,c instead of 2a,c: "The Dirac-like dispersion ... touching hole to electron pockets (Fig. 2 a, c) is the predicted hallmark" on page 8.

Author response: We thank the referee for bringing these points to our attention, we checked and fixed the remaining typos. We also specified that the core levels correspond to the 4f orbitals of Ta.

Fig. 3a is only meant to visually show the position of different cuts relative to the touching points, the values of the momenta can still be obtained from Fig. 2i, so we think the information presented is sufficient.

We changed 'three' with 'four' and fixed the reference to the figure.

Response to the comments by the reviewers, round 3

Reviewer's comment

The revised version of the manuscript has improved considerably through the detailed characterization by x-ray crystallography of the 2H-TaS₂ crystal from HQ Graphene studied by the authors. They report 11% 3R phase in the bulk 2H-TaS₂ crystal from their new XRD analysis. So, it is evident that a mixed-phase crystal has been studied.

Author's reply

We thank the referee for their detailed report and for acknowledging the improvements in the revised manuscript, in particular our detailed additional XRD analysis in our revised manuscript that revealed the presence of a 3R phase that we had earlier already detected with our ARPES measurements.

Reviewer's comment

The authors could not satisfactorily answer my previous comment, given below in italics: *“It is likely that the presence of the 3R phase in a 2H crystal, on which their whole study is based, depends on the growth conditions. So, this incomplete information indicates the possibility that, as I mentioned in my earlier report, the amount of the 3R phase in a different batch of crystals from different growers/suppliers might vary significantly. Thus, future researchers, who might want to work further on the present interesting findings reported here, will be put in a disadvantageous position. So, the authors should provide the details of the crystal growth. Alternately, the authors should show the existence of a similar amount of 3R phase at the surface (Fig. 2b) as obtained here on at least two crystals from other sources. This will show that the presence of the 3R phase is independent of any specific condition(s) of crystal growth.”* The reasons for my above comment, which is extremely relevant, are explained below:

The XRD study by the authors shows that the bulk 2H-TaS₂ crystal is multiphase with 11% 3R phase. The impression in the earlier version of the manuscript is that the 50% 3R phase occurs at the surface. But now it is clear that even the bulk crystal is multiphase. Importantly, this is in contradiction with the XRD, EDAX, and Raman studies on the 2H-TaS₂ crystals by HQ Graphene, where the 2H-TaS₂ crystals are shown to be >99.995% pure (see attachment below, reproduced from <https://www.hqgraphene.com/TaS2.php>). Moreover, the papers listed on this website, where a 2H-TaS₂ crystal from HQ Graphene was used, do not report evidence of the 3R phase.

Author's reply

We thank the reviewer for highlighting this point and for directing us to the HQ Graphene characterization. To clarify, the supplier's specification (>99.995 %) refers to elemental/chemical purity, most likely measured by EDX, rather than to structural phase purity. Standard XRD or Raman measurements as reported by HQ Graphene cannot determine polytype fractions with that level of precision, so the quoted figure must describe elemental composition. High elemental purity is fully compatible with small fractions of different stacking sequences (i.e., 2H/3R intergrowth), which are well documented to occur

during preparation of layered dichalcogenides samples (e.g., TaS₂: Chem. Eur. J. 23, 8082–8091 (2017); MoS₂: Cryst. Growth Des. 19, 5762–5767 (2019); NbS₂: Dalton Trans. 50, 3216–3223 (2021)).

Likewise, Raman spectroscopy does not reliably distinguish the 2H and 3R phases because the high-frequency intralayer phonon modes—those reported by HQ Graphene—are essentially identical for both stacking sequences (see, for example, ACS Nano 10, 1948–1953 (2016)). In addition, the single-crystal XRD plots provided by HQ Graphene display only the (00l) reflections, where the 2H and 3R peaks occur at nearly the same 2θ values; such data alone cannot rule out a minority 3R component.

On the other hand, our XRD measurements, using a much broader set of reflections along multiple directions and explicit structural refinement, resolve this ambiguity and clearly reveal a small 3R minority phase (~11%). We therefore see no contradiction with the HQ Graphene data. Rather, our analysis complements it by providing higher sensitivity to stacking order. We also note that several peaks in the cited single-crystal XRD pattern from HQ graphene show a systematic skewness toward lower 2θ , which would be expected from a minor 3R admixture, as the database entries show that the 3R phase is at slightly smaller angles than the 2H phase.

Reviewer's comment

The author's justification that "HQ Graphene does not disclose specific growth conditions—an unfortunate but common limitation when working with commercially sourced samples, which are now widely used for experimental studies in the condensed matter community" could possibly be acceptable if the crystals were pure single phase. However, in the case of complicated crystals and multiphase crystals, and in particular when the work fully depends on the impurity phase of the multiphase crystal, the details of crystal growth must be provided. In such cases, the norm in the scientific community is to include the crystal grower(s) as coauthor(s) of the manuscript. In this work, it is even more necessary because the multiphase nature of the 2H-TaS₂ crystal is in contradiction with the data shown by

crystal grower(s) (HQ Graphene).

Author's reply

We appreciate the reviewer's concern about transparency in sample sourcing and growth conditions. We agree that such details are valuable whenever they are available. In this case, however, our findings do not contradict the characterization provided by HQ Graphene. Their data and specifications address elemental purity, whereas our measurements reveal a small minority 3R stacking variant, something not excluded by the supplier's reports and well known to occur in layered dichalcogenides (references given above).

Our own XRD and ARPES analyses provide direct structural evidence for this multiphase nature, giving us confidence in the result even without specific growth details. We therefore believe the conclusions of our work remain well-supported and reproducible despite the absence of proprietary growth information from the commercial supplier.

Reviewer's comment

I suggested in my previous report that the authors should show the existence of a similar amount of 3R phase (as obtained here) on at least two crystals from other sources, in case they are unable to provide the details of crystal growth. In the absence of any details about crystal growth, this would have shown that their results are not dependent on a particular batch of crystals from HQ Graphene. The authors have failed to do that.

The above discussion shows that the foundation of the present work is rather weak, since it is done on the "impurity" phase (3R) of a multiphase crystal declared to be pure by the crystal grower (HQ Graphene).

Author's reply

We are grateful for the reviewer's suggestions and share the goal of ensuring that our results are robust and reproducible. However, we respectfully disagree with the view that our conclusions rest on a weak foundation simply because the 3R phase is a minority component of a commercially sourced crystal. Our central finding, the observation of a Kramers nodal-line metal with an Octdng and spindle-torus Fermi surface, is an intrinsic property of *any* 3R polytype itself. It does not depend on how that polytype is obtained or on its fraction within a given crystal.

We share the reviewer's commitment to reproducibility. Importantly, however, reproducibility does not depend on access to HQ Graphene's undisclosed growth conditions. What matters is the ability to identify the 3R phase, which can be done straightforwardly and conclusively by careful X-ray diffraction experiments, as we have demonstrated. Indeed, most of the HQ Graphene samples we examined contained measurable 3R fractions (see Supplementary Fig. 11).

The opportunity for other groups to reproduce our results is therefore not in question. Researchers may either obtain crystals directly from HQ Graphene and verify the presence of 3R inclusions by XRD, or intentionally grow phase-pure 3R-TaS₂ single crystals following established procedures in the literature (e.g., Gotoh et al., *J. Alloys Compd.* 270, 115 (1998)). Both routes provide a clear and reliable pathway to the 3R polytype whose intrinsic properties underpin our findings.

We appreciate the referee's suggestion to study samples from other suppliers. However, such measurements would not necessarily resolve the role of growth conditions, since uncontrolled

differences between suppliers could either suppress or enhance 3R fractions. We therefore believe that the most informative approach for our readers is to provide a prescription to thoroughly characterize the actual structure of the crystals used in our study, as we have done.

Reviewer's comment

An important implication of this disagreement is that the results of the earlier work in the literature (see the attached website where these works are listed) performed with HQ Graphene crystals on bulk 2H-TaS₂ crystals, which do not find evidence of the 3R phase, will be shown as a corollary of this work to be rather doubtful. This will create an undesirable situation for the research community working with commercial crystals.

Author's reply

We would like to reiterate that there is no disagreement between our findings and the published characterization of the supplier. Moreover, we believe our findings provide helpful context for the community by highlighting that commercially supplied crystals, even those advertised as highly pure, may contain small fractions of different structural phases. Recognizing this possibility should assist researchers in interpreting their own data and, in our view, ultimately strengthen rather than undermine future work with commercially sourced crystals.

Reviewer's comment

So, the authors must carefully scrutinize the history of the treatment done to the crystal(s) they have studied after these arrived in their laboratory. This will help them to pinpoint what led to the appearance of such a large 3R phase.

Author's reply

We thank the reviewer for this helpful suggestion. We have carefully examined the full handling history of our crystals and find no evidence of any post-growth treatment that could have induced the 3R domains. Our XRD refinements already reveal Ta interstitials within the 3R regions, providing a microscopic mechanism for their stabilization that is consistent with published reports. These observations indicate that the 3R phase formed during growth rather than in our laboratory.

To reflect this discussion, we have added a statement to our methods section:

"All crystals were used as received from HQ Graphene and were not subjected to any chemical, thermal, or mechanical treatment beyond standard handling for XRD and ARPES measurements. The presence of Ta interstitials revealed by our XRD refinements (see Supplementary Fig. 12) provides a plausible growth-related mechanism for stabilizing the 3R domains, consistent with previous literature [16]."

Reviewer's comment

At present, the authors are unable to provide any clue about how their crystals have 11% 3H phase in the bulk of a >99.995% pure 2H-TaS₂ crystal that they obtained from HQ Graphene.

This is essential for the scrutiny and reproducibility of the data presented here by the scientific community, which is the cornerstone of scientific research. However, since the authors were unable to address this important issue, manuscript cannot be recommended for acceptance in Nature Communications.

Author's reply

We would like to reiterate that the >99.995 % figure provided by HQ Graphene refers to elemental purity as determined by EDX, not to structural polytypism. Because the 2H and 3R phases share the same elemental composition, neither EDX nor standard Raman measurements can accurately distinguish them. Consequently, the supplier's specification is fully compatible with the minority 3R stacking sequence that we detect in our XRD and ARPES measurements.

We fully share the reviewer's concern for reproducibility, and we emphasize that the opportunity to reproduce our results is not in question. Our XRD refinements reveal Ta interstitials that provide a well-documented mechanism for stabilizing the 3R phase, consistent with prior literature. Moreover, established recipes for the intentional growth of phase-pure 3R-TaS₂ are available (e.g., Gotoh et al., J. Alloys Compd. 270, 115 (1998)), so other groups can either (i) obtain crystals from HQ Graphene and verify 3R inclusions by XRD, or (ii) grow phase-pure 3R crystals directly.

Most importantly, the central advance of our work, the observation of a Kramers nodal-line metal with an Octadong and spindle-torus Fermi surface, is an intrinsic property of the 3R polytype itself and does not depend on how the 3R phase is obtained. We therefore believe the scientific foundation of our study remains strong and that the results will be of broad value to the condensed-matter community.

Referee report on manuscript “Tunable Octadong and Spindle-Torus Fermi Surfaces in Kramers Nodal Line Metals” by G. Domaine et al.

In this work by G. Domaine et al., the authors reported tunable band structure with nontrivial topology in the 3R polytype of widely studied transition metal dichalcogenides (TMDCs). Specifically, the authors performed angle-resolved photoemission spectroscopy (ARPES) on TaS₂ and NbS₂. While 3R-NbS₂ crystals are directly available, 3R-TaS₂ phase resides in bulk crystals of 2H-TaS₂ and thus needs to be identified with microARPES spatial mapping.

Through ARPES measurement, as well as supportive DFT calculation, the authors claimed the observation of “Octadong” Fermi Surface (FS) in 3R-TaS₂ and “Spindle-Torus” FS in 3R-NbS₂, connected by a difference in chemical potential in these two systems. The quality of ARPES data is extraordinary. The theory is reasonably readable to non-expert audiences with some level of knowledge in topology. The manuscript is overall well-written. The observation and conclusion are interesting and insightful, and thus fit the scope of Nature Communications. However, before I can recommend this manuscript for publication in Nature Communications, there are some questions and suggestions that I hope the authors could address. Please see my comments below.

1. From the manuscript in its current form, it's hard to imagine and visualize the different crystal structures and stacking configurations. I strongly recommend the authors include a panel in Fig1 comparing the crystal structures of 1T, 1H, 2H and 3R polytypes of TMDCs
2. Where is the natural cleavage plane for the 3R polytype? From the crystal structure, theoretically there could be three different cleavage planes and thus three different surface terminations. The Fermi surfaces measured with ARPES largely depend on the surface termination type. Also, could there be surface reconstructions?
3. The 3R polytype has an out-of-plane lattice constant (c) of ~ 18 Angstrom. The ARPES measurements presented in this work are performed with UV photons with an energy around 50 eV. This is the most surface-sensitive regime in ARPES measurement, where, according to the universal curve of electron mean-free-path, the probing depth is < 4 Angstrom. This suggests that ARPES does not probe the entire unit cell in 3R polytype. Based on this factor, how shall we understand the ARPES spectrum?
4. TaS₂ systems in general host charge density wave (CDW) phases. Though the authors wisely chose a measurement temperature above the CDW long range order phase transition, the system (at least 2H polytype) hosts a short-range CDW at a much higher temperature (PHYSICAL REVIEW B 99,245144(2019)). The measurement temperature is very close to the transition temperature in 2H-TaS₂. In this temperature regime, one would naively expect a strong charge density fluctuation. Does this factor affect either ARPES measurement or DFT calculation?
5. The effect of matrix element on the ARPES spectrum is missing. Could the same features of the Fermi surfaces be observed in ARPES with different incident photon energy and polarization?

6. The proof of 3R polytype residing in 2H-TaS₂ is a bit handwavy. Stacking faults and misfit in TMDCs could also cause monolayer-like electronic structures (arXiv:2308.02772). The authors attempted to differentiate the single-band FS structure from that of monolayer 1H phase from an electronic perspective. However, the difference among distinct polytypes lies in the crystal structure after all. Thus, a structural probe is the smoking-gun evidence of a such phase mixing. Thus, I suggest additional measurements that reflect the distinct crystal structures in different domains, ideally on the same sample where ARPES was measured. Any of nano X-ray diffraction, spatially resolved Raman scattering and transmission electron microscopy should in principle be sufficient.

I am willing to reconsider this manuscript for publication after the questions above are properly addressed.

The revised version of the manuscript has improved considerably through the detailed characterization by x-ray crystallography of the 2H-TaS₂ crystal from HQ Graphene studied by the authors. They report 11% 3R phase in the bulk 2H-TaS₂ crystal from their new XRD analysis. So, it is evident that a mixed-phase crystal has been studied.

The authors could not satisfactorily answer my previous comment, given below in italics: *“It is likely that the presence of the 3R phase in a 2H crystal, on which their whole study is based, depends on the growth conditions. So, this incomplete information indicates the possibility that, as I mentioned in my earlier report, the amount of the 3R phase in a different batch of crystals from different growers/suppliers might vary significantly. Thus, future researchers, who might want to work further on the present interesting findings reported here, will be put in a disadvantageous position. So, the authors should provide the details of the crystal growth. Alternately, the authors should show the existence of a similar amount of 3R phase at the surface (Fig. 2b) as obtained here on at least two crystals from other sources. This will show that the presence of the 3R phase is independent of any specific condition(s) of crystal growth.”* The reasons for my above comment, which is extremely relevant, are explained below:

The XRD study by the authors shows that the bulk 2H-TaS₂ crystal is multiphase with 11% 3R phase. The impression in the earlier version of the manuscript is that the 50% 3R phase occurs at the surface. But now it is clear that even the bulk crystal is multiphase. Importantly, this is in contradiction with the XRD, EDAX, and Raman studies on the 2H-TaS₂ crystals by HQ Graphene, where the 2H-TaS₂ crystals are shown to be >99.995% pure (see attachment below, reproduced from <https://www.hqgraphene.com/TaS2.php>). Moreover, the papers listed on this website, where a 2H-TaS₂ crystal from HQ Graphene was used, do not report evidence of the 3R phase.

The author's justification that *“HQ Graphene does not disclose specific growth conditions—an unfortunate but common limitation when working with commercially sourced samples, which are now widely used for experimental studies in the condensed matter community”* could possibly be acceptable if the crystals were pure single phase. However, in the case of complicated crystals and multiphase crystals, and in particular when the work fully depends on the impurity phase of the multiphase crystal, the details of crystal growth must be provided. In such cases, the norm in the scientific community is to include the crystal grower(s) as coauthor(s) of the manuscript. In this work, it is even more necessary because the multiphase nature of the 2H-TaS₂ crystal is in contradiction with the data shown by crystal grower(s) (HQ Graphene).

I suggested in my previous report that the authors should show the existence of a similar amount of 3R phase (as obtained here) on at least two crystals from other sources, in case they are unable to provide the details of crystal growth. In the absence of any details about crystal growth, this would have shown that their results are not dependent on a particular batch of crystals from HQ Graphene. The authors have failed to do that.

The above discussion shows that the foundation of the present work is rather weak, since it is done on the “impurity” phase (3R) of a multiphase crystal declared to be pure by the crystal grower (HQ Graphene). An important implication of this disagreement is that the results of the earlier work in the literature (see the attached website where these works are listed) performed with HQ Graphene crystals on bulk 2H-TaS₂ crystals, which do not find evidence of the 3R phase, will be shown as a corollary of this work to be rather doubtful.

This will create an undesirable situation for the research community working with commercial crystals.

So, the authors must carefully scrutinize the history of the treatment done to the crystal(s) they have studied after these arrived in their laboratory. This will help them to pinpoint what led to the appearance of such a large 3R phase.

At present, the authors are unable to provide any clue about how their crystals have 11% 3H phase in the bulk of a >99.995% pure 2H-TaS₂ crystal that they obtained from HQ Graphene. This is essential for the scrutiny and reproducibility of the data presented here by the scientific community, which is the cornerstone of scientific research. However, since the authors were unable to address this important issue, manuscript cannot be recommended for acceptance in Nature Communications.